# Optimising Physics-Informed Neural Network Solvers for Turbulence Modelling: A Study on Solver Constraints Against a Data-Driven Approach

William Fox, Bharath Sharma , Jianhua Chen, Marco Castellani and Daniel M. Espino *

Department of Mechanical Engineering, University of Birmingham, Birmingham B15 2TT, UK;
m.castellani@bham.ac.uk (M.C.)
* Correspondence: d.m.espino@bham.ac.uk

**Abstract:** Physics-informed neural networks (PINNs) have emerged as a promising approach for simulating nonlinear physical systems, particularly in the field of fluid dynamics and turbulence modelling. Traditional turbulence models often rely on simplifying assumptions or closed numerical models, which simplify the flow, leading to inaccurate flow predictions or long solve times. This study examines solver constraints in a PINNs solver, aiming to generate an understanding of an optimal PINNs solver with reduced constraints compared with the numerically closed models used in traditional computational fluid dynamics (CFD). PINNs were implemented in a periodic hill flow case and compared with a simple data-driven approach to neural network modelling to show the limitations of a data-driven model on a small dataset (as is common in engineering design). A standard full equation PINNs model with predicted first-order stress terms was compared against reduced-boundary models and reduced-order models, with different levels of assumptions made about the flow to monitor the effect on the flow field predictions. The results in all cases showed good agreement against direct numerical simulation (DNS) data, with only boundary conditions provided for training as in numerical modelling. The efficacy of reduced-order models was shown using a continuity only model to accurately predict the flow fields within 0.147 and 2.6 percentage errors for streamwise and transverse velocities, respectively, and a modified mixing length model was used to show the effect of poor assumptions on the model, including poor convergence at the flow boundaries, despite a reduced solve time compared with a numerically closed equation set. The results agree with contemporary literature, indicating that physics-informed neural networks are a significant improvement in solve time compared with a data-driven approach, with a novel proposition of numerically derived unclosed equation sets being a good representation of a turbulent system. In conclusion, it is shown that numerically unclosed systems can be efficiently solved using reduced-order equation sets, potentially leading to a reduced compute requirement compared with traditional solver methods.

**Keywords:** data-driven model; flow predictions; numerically closed; periodic hill; physics-informed neural network; turbulence modelling



## 1. Introduction

Flow modelling is an important part of the engineering design process, allowing for optimisation of the aerodynamics of a design, design validation for hydraulic or other fluid transporting components/mechanisms, and safety and power validation of complex designs including aircraft turbine engines. In many of these engineering applications, turbulent flow is a common physical phenomenon to be accounted for during flow modelling for design validation. Generally, only averaged flow calculations are performed due to the prohibitive cost of a transient numerical solver [1].

Two promising approaches for turbulent flow modelling include utilisation of neural networks, specifically data-driven modelling and a physics-informed modelling approach.



A data-driven approach uses data from previous numerical solvers to create a generalised map of the nonlinear dynamics between the input geometry and the output flow fields [2] for a large set of training data. The benefit of this approach is an almost instant solve time, after an initial training phase, and provided that the model is generalisable the same model can be used to predict on several different geometries.

Physics-informed neural networks (PINNs) are a predictive solving method for per-geometry solutions, like CFD, where geometry and boundary conditions are defined rather than training on a large dataset. They are therefore not governed by comparing predictions to data, but rather to a governing set of equations. PINNs have developed from simple modelling of ODEs and PDEs [3], to solving more complex PDE systems such as the Schrodinger and Burgers equations [4]. PINNs have been used to converge using the Navier–Stokes equations to accurately solve flow cases including cylinder flow [5]. Current research involves modelling of turbulent flow, including the Falkner–Skan boundary with adverse pressure gradient, APG turbulent boundary layer, and a periodic hill [6], giving promising results for streamwise and transverse velocity predictions. Several governing equation sets were also tested on a backwards-facing step [7]; however, predictions only including boundary conditions were poor, requiring additional data to be fed to the neural network to converge to the correct solution.

In all these implementations, a mostly numerically closed system has been used to govern the loss function; however, a benefit of the backpropagation method used in a neural network architecture compared with traditional solvers is that the system does not need to be fully defined to provide a solution. This potentially allows for the equation set to be either simplified, or boundary data to be removed, to allow for a faster solve time or a more flexible solving method than a traditional numerical solver.

Reduced-order modelling (ROM) is a method used to reduce the complexity of a system of equations while still remaining representative of the dynamics of the system. It has been successfully implemented in problems such as the Helmholtz decomposition to reduce the complexity of the system by reducing the number of components to solve [8]. Reduced-order modelling (ROM) has also previously been investigated using PINNs with a data-driven component by Fu et al. [9], where an interesting approach to generating ROMs is used with snapshot data. This method is highly versatile and computationally efficient, but this generation relies on an understanding of either the geometry or having snapshot data to use in the data-driven component.

A very interesting application that will not be explored in this paper is the generation of ROMs using neural networks. It was shown by Brunton et al. [10] that neural networks are capable of parametrising parametric systems successfully and simply as the system may be traditionally defined. Neural networks have also been successfully used to reduce the dimensionality of a governing equation to simplify the characteristic function [11]. Finding a unique and simple characteristic equation for a given system using neural networks could be an interesting separate topic, but this paper will focus on a numerical approach to generating a ROM for the physical system of equations. This is to simplify the scope and ensure that results are directly comparable to numerical solutions that could be easily generated using commercial numerical solvers.

Contributions listed have been very focused on reduced-order modelling and PINNs, but a wide range of uses for PINNs have been outlined by Vinuesa and Brunton [12], including accelerating DNS models and LES modelling. ROMs are also discussed, including autoencoders, which generate a reduced coordinate system to represent a reduced dynamic system.

The aim was to analyse reduced-order models in several configurations, including a full numerically closed model, reduced-boundary-enforced model(s), a reduced-equation model, a numerically closed turbulence model with assumptions, and the same turbulence model with the least accurate assumptions removed, generated with no previous understanding of geometry or snapshots of data, only from boundary conditions similar to a numerical solver. The analysis shows the effect on the solutions and solve time, and com-

pares this against a simple implementation of a data-driven approach. The motivation for this topic was to provide recommendations for solver inputs and equations for simple 2D incompressible flow cases, along with benchmark data for several turbulence models run using PINNs on a widely available dataset, with a direct data-driven comparable model.

*Formulation of the Problem*

A difficulty in turbulence modelling for flow simulations comes from the calculation and approximation of the first-order stress terms, often referred to as turbulence closure or the closure problem. The 2D incompressible Reynolds-averaged Navier–Stokes (RANS) equations are given,

$$\frac{\partial \bar{u}}{\partial x} + \frac{\partial \bar{v}}{\partial y} = 0 \tag{1}$$

$$\bar{u}\frac{\partial \bar{u}}{\partial x} + \bar{v}\frac{\partial \bar{u}}{\partial y} - \left(-\frac{1}{\rho}\right)\frac{\partial \bar{p}}{\partial x} - \nu\left(\frac{\partial^2 \bar{u}}{\partial x^2} + \frac{\partial^2 \bar{u}}{\partial y^2}\right) + \overline{v'\frac{\partial u'}{\partial y}} + \overline{u'\frac{\partial u'}{\partial x}} = 0 \tag{2}$$

$$\bar{u}\frac{\partial \bar{v}}{\partial x} + \bar{v}\frac{\partial \bar{v}}{\partial y} - \left(-\frac{1}{\rho}\right)\frac{\partial \bar{p}}{\partial y} - \nu\left(\frac{\partial^2 \bar{v}}{\partial x^2} + \frac{\partial^2 \bar{v}}{\partial y^2}\right) + \overline{v'\frac{\partial v'}{\partial y}} + \overline{u'\frac{\partial v'}{\partial x}} = 0 \tag{3}$$

where $\bar{u}$ is the averaged flow velocity in the streamwise (x) direction, $\bar{v}$ is the averaged flow velocity in the transverse (y) direction, $\rho$ is fluid density, $\bar{p}$ is the averaged pressure, $\nu$ is the fluid kinematic viscosity, and the prime represents fluctuating flow components. For the full derivations, see Appendix A. The first-order stress terms are given,

$$\frac{\partial \tau'_{xx}}{\partial x} + \frac{\partial \tau'_{xy}}{\partial y} = \overline{v'\frac{\partial u'}{\partial y}} + \overline{u'\frac{\partial u'}{\partial x}} \tag{4}$$

$$\frac{\partial \tau'_{yx}}{\partial x} + \frac{\partial \tau'_{yy}}{\partial y} = \overline{v'\frac{\partial v'}{\partial y}} + \overline{u'\frac{\partial v'}{\partial x}} \tag{5}$$

It should also be noted that, since the flow is incompressible in this case,

$$\frac{\partial \tau'_{xy}}{\partial y} = \frac{\partial \tau'_{yx}}{\partial x} \tag{6}$$

This can be physically interpreted as the lack of a preferred direction in the turbulent fluctuations of an incompressible flow. The turbulent fluctuations are equally likely to cause a shear stress in the x-y and y-x planes, as the flow is isotropic.

The first-order stress terms are not made up of averaged flow components, but fluctuating values of the velocity in the streamwise and transverse directions. Since the RANS equations are arranged to solve flow in terms of averaged flow quantities, these quantities must be either related to the mean flow quantities (without relying on time series data, which in this case is not available) or predicted separately as a function of geometry or other physical domain features using a turbulence closure model. This is often inaccurate when using simple models such as mixing length [13], or computationally inefficient using complex models such as RSM [14].

## 2. Periodic Hill

A periodic hill flow condition was chosen as the CFD benchmark case. It is commonly used in CFD benchmarks [15–17], partially due to well-defined geometry, simplifying setup. Additionally, the presence of rich turbulent phenomena including vortex shedding, flow separation, and reattachment and transition regions challenges traditional CFD solvers including RANS models. Due to their common use as a turbulence modelling benchmark, there are a large number of CFD data available, which are required to train a data-driven model. The dataset selected is provided by Xiao et al. [18], and the solver used is the high-order flow solver Incompact3d. This solver uses a sixth-order compact finite differ-

ence scheme on a monobloc Cartesian mesh. The source code for this solver is publicly available on GitHub [19]. From recent literature, this dataset provides the most detail on validation and solving, and was developed with data-driven models in mind, making it a suitable dataset to compare data-driven and nonlinear dynamics-informed approaches to modelling turbulence.

The DNS solve the forced incompressible Navier–Stokes equations for a fluid field. The equations are given in [18] as,

$$\nabla \cdot u = 0 \tag{7}$$

$$\frac{\partial u}{\partial t} = -\frac{1}{\rho}\nabla p - \frac{1}{2}[\nabla(u \otimes u) + (u \cdot \nabla)u] + \nu\nabla^2 u + f \tag{8}$$

where $p$ is the pressure field, $u$ is the velocity field, $x$ is the spatial domain (from div operators), $t$ is the temporal domain, and $\otimes$ denotes the vector outer-product. In this equation set, the forcing term $f$ is used to enforce a specified mass flux at the boundary through the immersed boundary method. Incompressible here indicates that spatial variation of the fluid density is ignored in these equations as a simplification. Validation of the code can be found in [20].

It is important to note that, although the data provided from these equations are a steady-state flow case, they are not run using the RANS equations, which average the velocity and pressure fields over a discrete time. These data are provided by running a transient solver to a point of steady-state flow, which the RANS equations would give as in this dataset, and the steady-state flow condition is maintained over the observed time.

The dataset contains 29 different periodic hill flow geometries for a Reynold's number of the flow of 5600. The geometries are parameterised by a set of third-order equations (Appendix B). A subset of the geometries are shown in Figure 1.

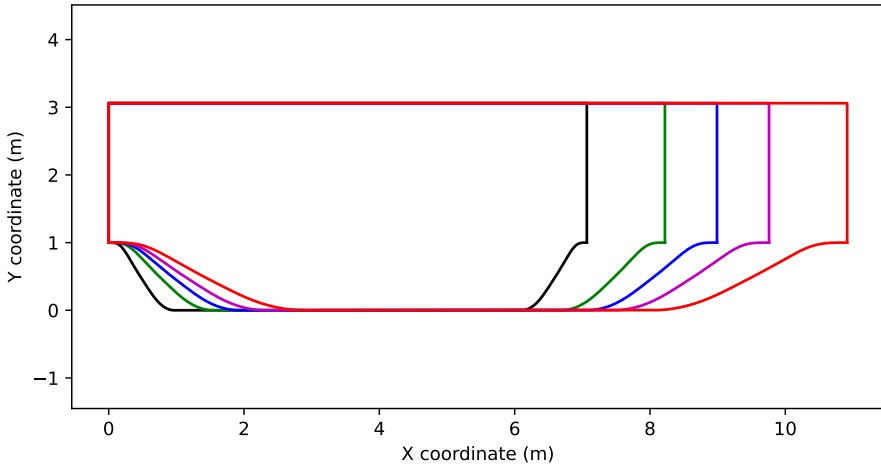

**Figure 1.** Geometries varying a parameter defining the slope characteristics, $\alpha$, and a parameter defining the domain length, $l$. Five of 25 examples shown, varying slope and domain length. More information provided by [19].

## 3. Methods

### 3.1. Data-Driven Model

#### 3.1.1. Network Architecture

The architecture of the data-driven model is a simple dense multilayer perceptron (MLP) network, consisting of 9 hidden layers of varying node numbers, from 128 reduced to 8 nodes to improve the consistency of convergence of the model (shown in Table 1). This was chosen as the most basic data-driven architecture for time-averaged data, to give a basic benchmark against PINNs models, and highlight areas for improvement in this type of modelling should a data-driven approach be taken. The input to the neural network is a

tensor containing six independent variables for a given nodal position in any of the training dataset. These variables include the $x$ and $y$ positions of the node, the calculated distance, $d$, between the node and the bottom wall, and three geometric parameters to describe the periodic hill geometry. These are hill slope parameter, $\alpha$, domain length, $l$, and domain height, $h$, to help characterise the overall domain geometry for the nodal predictions. The output tensor of the neural network is six dependant variables, including the time-averaged streamwise velocity, $\bar{u}$, the time-averaged transverse velocity, $\bar{v}$, and the time-averaged pressure, $\bar{p}$, and the first-order Reynold's stress terms $u'u'$, $u'v'$, and $v'v'$.

For completeness, note that Table 2 presents the architecture for the data driven model, Table 3 outlines the computational settings, Table 4 includes a summary of the PINNs models implemented, and data driven metrics are presented in Table 5. The full model constraints for all models can be seen in Table 6.

**Table 1.** Data-driven neural network architecture.

| Layer Type | Output Shape | Activation Function | Batch Normalisation |
|---|---|---|---|
| Input Layer | (6,) | – | – |
| Dense | (128,) | ReLU | Yes |
| Dense | (64,) | ReLU | Yes |
| Dense | (64,) | ReLU | Yes |
| Dense | (32,) | ReLU | Yes |
| Dense | (32,) | ReLU | Yes |
| Dense | (16,) | ReLU | Yes |
| Dense | (16,) | ReLU | Yes |
| Dense | (8,) | ReLU | Yes |
| Dense | (8,) | ReLU | Yes |
| Output Layer | (6,) | – | – |

**Table 2.** Data-driven neural network architecture.

| Layer Type | Output Shape | Activation Function | Batch Normalisation |
|---|---|---|---|
| Input Layer | (input shape,) | – | – |
| Dense | (20,) | tanh | No |
| Dense | (20,) | tanh | No |
| Dense | (20,) | tanh | No |
| Dense | (20,) | tanh | No |
| Dense | (20,) | tanh | No |
| Dense | (20,) | tanh | No |
| Dense | (20,) | tanh | No |
| Dense | (20,) | tanh | No |
| Output Layer | (output shape,) | – | – |

**Table 3.** Computational settings.

| Computer | AORUS 15P XD Laptop (Singapore, Rep. Singapore) |
|---|---|
| Processor | 11th Gen Intel ® Core i7-11800H (Intel, Santa Clara, CA, USA) |
| RAM | 32 GB |
| OS | Windows 11 23H2 |
| Modules | TensorFlow 2.14.0<br>pyDOE 0.3.8<br>keras 2.14.0<br>numpy 1.26.0<br>matplotlib 3.8.0<br>scipy 1.11.3 |
| Instruction Set | CPU |

**Table 4.** PINNs methods summary.

| Method Name | Neural Network Inputs | Neural Network Outputs | Governing Flow Equations | Enforced Boundaries |
|---|---|---|---|---|
| Direct Reynolds Stress | $\begin{pmatrix} x \\ y \end{pmatrix}$ | $\begin{pmatrix} \overline{u} \\ \overline{v} \\ \overline{p} \\ u'u' \\ u'v' \\ v'v' \end{pmatrix}$ | 2D incompressible Continuity, Momentum in x, Momentum in y <br> $\frac{\partial \overline{u}}{\partial x} + \frac{\partial \overline{v}}{\partial y} = 0$ <br> $\overline{u}\frac{\partial \overline{u}}{\partial x} + \overline{v}\frac{\partial \overline{u}}{\partial y} - \left(-\frac{1}{\rho}\right)\frac{\partial \overline{p}}{\partial x} - \nu\left(\frac{\partial^2 \overline{u}}{\partial x^2} + \frac{\partial^2 \overline{u}}{\partial y^2}\right) + \overline{v'\frac{\partial u'}{\partial y}} + \overline{u'\frac{\partial u'}{\partial x}} = 0$ <br> $\overline{u}\frac{\partial \overline{v}}{\partial x} + \overline{v}\frac{\partial \overline{v}}{\partial y} - \left(-\frac{1}{\rho}\right)\frac{\partial \overline{p}}{\partial y} - \nu\left(\frac{\partial^2 \overline{v}}{\partial x^2} + \frac{\partial^2 \overline{v}}{\partial y^2}\right) + \overline{v'\frac{\partial v'}{\partial y}} + \overline{u'\frac{\partial v'}{\partial x}} = 0$ | $\begin{pmatrix} \overline{u} \\ \overline{v} \\ \overline{p} \\ u'u' \\ u'v' \\ v'v' \end{pmatrix}$ |
| Direct Reynolds Stress with Reduced Boundary Enforcement | $\begin{pmatrix} x \\ y \end{pmatrix}$ | $\begin{pmatrix} \overline{u} \\ \overline{v} \\ \overline{p} \\ u'u' \\ u'v' \\ v'v' \end{pmatrix}$ | 2D incompressible Continuity, Momentum in x, Momentum in y <br> $\frac{\partial \overline{u}}{\partial x} + \frac{\partial \overline{v}}{\partial y} = 0$ <br> $\overline{u}\frac{\partial \overline{u}}{\partial x} + \overline{v}\frac{\partial \overline{u}}{\partial y} - \left(-\frac{1}{\rho}\right)\frac{\partial \overline{p}}{\partial x} - \nu\left(\frac{\partial^2 \overline{u}}{\partial x^2} + \frac{\partial^2 \overline{u}}{\partial y^2}\right) + \overline{v'\frac{\partial u'}{\partial y}} + \overline{u'\frac{\partial u'}{\partial x}} = 0$ <br> $\overline{u}\frac{\partial \overline{v}}{\partial x} + \overline{v}\frac{\partial \overline{v}}{\partial y} - \left(-\frac{1}{\rho}\right)\frac{\partial \overline{p}}{\partial y} - \nu\left(\frac{\partial^2 \overline{v}}{\partial x^2} + \frac{\partial^2 \overline{v}}{\partial y^2}\right) + \overline{v'\frac{\partial v'}{\partial y}} + \overline{u'\frac{\partial v'}{\partial x}} = 0$ | $\begin{pmatrix} \overline{u} \\ \overline{v} \\ \overline{p} \end{pmatrix}$ <br> $\begin{pmatrix} \overline{u} \\ \overline{v} \end{pmatrix}$ |
| Continuity Only Model | $\begin{pmatrix} x \\ y \end{pmatrix}$ | $\begin{pmatrix} \overline{u} \\ \overline{v} \end{pmatrix}$ | 2D incompressible Continuity <br> $\frac{\partial \overline{u}}{\partial x} + \frac{\partial \overline{v}}{\partial y} = 0$ | $\begin{pmatrix} \overline{u} \\ \overline{v} \end{pmatrix}$ |
| Mixing Length | $\begin{pmatrix} x \\ y \\ d \end{pmatrix}$ | $\begin{pmatrix} \overline{u} \\ \overline{v} \\ \overline{p} \end{pmatrix}$ | 2D incompressible Continuity, Momentum in x, Momentum in y, First-order Stresses, Turbulent Viscosity, Mixing Length, Strain Tensor <br> $\frac{\partial \overline{u}}{\partial x} + \frac{\partial \overline{v}}{\partial y} = 0$ <br> $\overline{u}\frac{\partial \overline{u}}{\partial x} + \overline{v}\frac{\partial \overline{u}}{\partial y} - \left(-\frac{1}{\rho}\right)\frac{\partial \overline{p}}{\partial x} - \nu\left(\frac{\partial^2 \overline{u}}{\partial x^2} + \frac{\partial^2 \overline{u}}{\partial y^2}\right) + \overline{v'\frac{\partial u'}{\partial y}} + \overline{u'\frac{\partial u'}{\partial x}} = 0$ <br> $\overline{u}\frac{\partial \overline{v}}{\partial x} + \overline{v}\frac{\partial \overline{v}}{\partial y} - \left(-\frac{1}{\rho}\right)\frac{\partial \overline{p}}{\partial y} - \nu\left(\frac{\partial^2 \overline{v}}{\partial x^2} + \frac{\partial^2 \overline{v}}{\partial y^2}\right) + \overline{v'\frac{\partial v'}{\partial y}} + \overline{u'\frac{\partial v'}{\partial x}} = 0 \quad \overline{\tau_{xx}} = 2\mu_t\left(\frac{\partial \overline{u}}{\partial x} + \frac{\partial \overline{u}}{\partial x} + \frac{\partial \overline{v}}{\partial y}\right)$ <br> $\overline{\tau_{yy}} = 2\mu_t\left(\frac{\partial \overline{u}}{\partial x} + \frac{\partial \overline{v}}{\partial y} + \frac{\partial \overline{v}}{\partial y}\right)$ <br> $\overline{\tau_{xy}} = \overline{\tau_{yx}} = \mu_t\left(\frac{\partial \overline{u}}{\partial x} + \frac{\partial \overline{v}}{\partial y}\right) \quad \mu_t = l_m^2 \sqrt{G}$ <br> $l_m = min(0.419d, 0.09d_{max})$ <br> $G = 2\left(\frac{\partial \overline{u}}{\partial x}\right)^2 + 2\left(\frac{\partial \overline{v}}{\partial y}\right)^2 + \left(\frac{\partial \overline{u}}{\partial y} + \frac{\partial \overline{v}}{\partial x}\right)^2$ | $\begin{pmatrix} \overline{u} \\ \overline{v} \\ \overline{p} \end{pmatrix}$ |
| Turbulent Viscosity | $\begin{pmatrix} x \\ y \end{pmatrix}$ | $\begin{pmatrix} \overline{u} \\ \overline{v} \\ \overline{p} \\ \mu_t \end{pmatrix}$ | 2D incompressible Continuity, Momentum in x, Momentum in y, First-order Stresses, Turbulent Viscosity, Mixing Length, Strain Tensor <br> $\frac{\partial \overline{u}}{\partial x} + \frac{\partial \overline{v}}{\partial y} = 0$ <br> $\overline{u}\frac{\partial \overline{u}}{\partial x} + \overline{v}\frac{\partial \overline{u}}{\partial y} - \left(-\frac{1}{\rho}\right)\frac{\partial \overline{p}}{\partial x} - \nu\left(\frac{\partial^2 \overline{u}}{\partial x^2} + \frac{\partial^2 \overline{u}}{\partial y^2}\right) + \overline{v'\frac{\partial u'}{\partial y}} + \overline{u'\frac{\partial u'}{\partial x}} = 0$ <br> $\overline{u}\frac{\partial \overline{v}}{\partial x} + \overline{v}\frac{\partial \overline{v}}{\partial y} - \left(-\frac{1}{\rho}\right)\frac{\partial \overline{p}}{\partial y} - \nu\left(\frac{\partial^2 \overline{v}}{\partial x^2} + \frac{\partial^2 \overline{v}}{\partial y^2}\right) + \overline{v'\frac{\partial v'}{\partial y}} + \overline{u'\frac{\partial v'}{\partial x}} = 0 \quad \overline{\tau_{xx}} = 2\mu_t\left(\frac{\partial \overline{u}}{\partial x} + \frac{\partial \overline{u}}{\partial x} + \frac{\partial \overline{v}}{\partial y}\right)$ <br> $\overline{\tau_{yy}} = 2\mu_t\left(\frac{\partial \overline{u}}{\partial x} + \frac{\partial \overline{v}}{\partial y} + \frac{\partial \overline{v}}{\partial y}\right)$ <br> $\overline{\tau_{xy}} = \overline{\tau_{yx}} = \mu_t\left(\frac{\partial \overline{u}}{\partial x} + \frac{\partial \overline{v}}{\partial y}\right)$ | $\begin{pmatrix} \overline{u} \\ \overline{v} \\ \overline{p} \end{pmatrix}$ |
| Turbulent Viscosity and Turbulent Kinetic Energy | $\begin{pmatrix} x \\ y \end{pmatrix}$ | $\begin{pmatrix} \overline{u} \\ \overline{v} \\ \overline{p} \\ \mu_t \end{pmatrix}$ | 2D incompressible Continuity, Momentum in x, Momentum in y, First-order Stresses, Turbulent Viscosity, Mixing Length, Strain Tensor <br> $\frac{\partial \overline{u}}{\partial x} + \frac{\partial \overline{v}}{\partial y} = 0$ <br> $\overline{u}\frac{\partial \overline{u}}{\partial x} + \overline{v}\frac{\partial \overline{u}}{\partial y} - \left(-\frac{1}{\rho}\right)\frac{\partial \overline{p}}{\partial x} - \nu\left(\frac{\partial^2 \overline{u}}{\partial x^2} + \frac{\partial^2 \overline{u}}{\partial y^2}\right) + \overline{v'\frac{\partial u'}{\partial y}} + \overline{u'\frac{\partial u'}{\partial x}} = 0$ <br> $\overline{u}\frac{\partial \overline{v}}{\partial x} + \overline{v}\frac{\partial \overline{v}}{\partial y} - \left(-\frac{1}{\rho}\right)\frac{\partial \overline{p}}{\partial y} - \nu\left(\frac{\partial^2 \overline{v}}{\partial x^2} + \frac{\partial^2 \overline{v}}{\partial y^2}\right) + \overline{v'\frac{\partial v'}{\partial y}} + \overline{u'\frac{\partial v'}{\partial x}} = 0 \quad \overline{\tau_{xx}} =$ <br> $\mu_t\left(\frac{\partial \overline{u}}{\partial x} + \frac{\partial \overline{u}}{\partial x}\right) - \frac{2}{3}\rho k$ <br> $\overline{\tau_{yy}} = \mu_t\left(\frac{\partial \overline{u}}{\partial y} + \frac{\partial \overline{v}}{\partial x}\right)$ <br> $\overline{\tau_{xx}} = \mu_t\left(\frac{\partial \overline{v}}{\partial y} + \frac{\partial \overline{v}}{\partial y}\right) - \frac{2}{3}\rho k$ | $\begin{pmatrix} \overline{u} \\ \overline{v} \\ \overline{p} \end{pmatrix}$ |

**Table 5.** Data-driven metrics for all dataset geometries. Runs 2 and 3 can be found in Appendix F *.

| Slope Parameter | Domain Length (m) | Domain Height (m) | Solve Time (hrs) | Epochs to Converge | $\overline{u}$ Error (%) | $\overline{v}$ Error (%) | $\overline{p}$ Error (%) |
|---|---|---|---|---|---|---|---|
| 5 | 10.0710 | 2.0240 | 4.14 | 64 | 10.7 | 53.6 | 107.4 |
| 5 | 10.0710 | 3.0360 | 4.30 | 71 | 3.8 | 25.3 | 109.1 |
| 5 | 10.0710 | 4.0480 | 6.43 | 100 | 2.4 | 23.7 | 117.2 |
| 5 | 4.0710 | 2.0240 | 6.57 | 100 | 9.1 | 69.3 | 157.6 |
| 5 | 4.0710 | 3.0360 | 3.14 | 51 | 2.8 | 67.5 | 215.9 |
| 5 | 4.0710 | 4.0480 | 6.40 | 98 | 2.9 | 60.7 | 253.9 |
| 5 | 7.0710 | 2.0240 | 6.15 | 100 | 5.3 | 39.6 | 110.4 |
| 5 | 7.0710 | 3.0360 | 6.07 | 100 | 2.9 | 30.0 | 123.3 |
| 5 | 7.0710 | 4.0480 | 6.00 | 98 | 2.2 | 23.8 | 135.1 |

**Table 5.** *Cont.*

| Slope Parameter | Domain Length (m) | Domain Height (m) | Solve Time (hrs) | Epochs to Converge | $\bar{u}$ Error (%) | $\bar{v}$ Error (%) | $\bar{p}$ Error (%) |
|---|---|---|---|---|---|---|---|
| 7.5 | 8.0355 | 3.0360 | 2.16 | 35 | 19.2 | 127.6 | 130.5 |
| 10 | 12.0000 | 2.0240 | 4.07 | 62 | 8226.0 | 52,713.0 | 7328.0 |
| 10 | 12.0000 | 3.0360 | 3.36 | 55 | 19.3 | 77.7 | 113.0 |
| 10 | 12.0000 | 4.0480 | 3.91 | 62 | 18.4 | 103.6 | 114.7 |
| 10 | 6.0000 | 2.0240 | 1.98 | 31 | 17.4 | 117.0 | 118.2 |
| 10 | 6.0000 | 3.0360 | 4.14 | 67 | 9.8 | 95.4 | 147.0 |
| 10 | 6.0000 | 4.0480 | 2.56 | 40 | 10.0 | 146.4 | 174.7 |
| 10 | 9.0000 | 2.0240 | 2.97 | 46 | 21.0 | 81.3 | 106.6 |
| 10 | 9.0000 | 3.0360 | 2.11 | 35 | 14.8 | 90.0 | 114.9 |
| 10 | 9.0000 | 4.0480 | 3.65 | 59 | 10.6 | 83.4 | 119.9 |
| 12.5 | 9.9645 | 3.0360 | 6.13 | 100 | 1169.5 | 1431.7 | 1372.3 |
| 15 | 10.9090 | 2.0240 | 4.59 | 74 | 12.0 | 39.2 | 103.6 |
| 15 | 10.9090 | 3.0360 | 1.89 | 31 | 18.4 | 87.3 | 106.3 |
| 15 | 10.9090 | 4.0480 | 2.40 | 40 | 5.5 | 30.2 | 106.6 |
| 15 | 10.9090 | 2.0240 | 3.70 | 59 | 31.4 | 109.9 | 108.9 |
| 15 | 13.9290 | 3.0360 | 2.90 | 48 | 25.2 | 95.5 | 108.4 |
| 15 | 13.9290 | 4.0480 | 1.94 | 31 | 30.7 | 103.2 | 110.5 |
| 15 | 7.9290 | 2.0240 | 2.22 | 34 | 25.4 | 98.6 | 104.4 |
| 15 | 7.9290 | 3.0360 | 2.39 | 39 | 17.7 | 97.9 | 111.8 |
| 15 | 7.9290 | 4.0480 | 3.93 | 61 | 14.0 | 110.3 | 119.5 |
| **Median Average** | | | **3.70 ± 1.51** | **59 ± 25** | **12.0 ± 8.73** | **83.4 ± 34.5** | **114.7 ± 35.6** |

\* See Appendix E for calculation method.

**Table 6.** Solver metrics and overall accuracy.

| Neural Network Name | Neural Network Abbreviation | Training Time (hrs) | Prediction Time | $\bar{u}$ Error (%) * | $\bar{v}$ Error (%) * | $\bar{p}$ Error (%) * | Description |
|---|---|---|---|---|---|---|---|
| Dense MLP Data Driven | Data Driven | 3.14 | 11.0 | 2.85 | 67.512 | 215.897 | See Section 3.1 |
| Direct Reynold's Stress Model | DRSM | 1.61 | - | 0.613 | 9.244 | 4.78 | See Section 3.2.4 |
| Direct Reynold's Stress Model—Reduced Boundary Enforcement, $u, v, p$ Enforcement | $\text{RBM}_{uvp}$ | 1.36 | - | 0.309 | 3.487 | 5.783 | See Section 3.2.5 |
| Direct Reynold's Stress Model—Reduced Boundary Enforcement, $u, v$ enforcement | $\text{RBM}_{uv}$ | 1.82 | - | 0.233 | 3.28 | 253.675 | See Section 3.2.5 |
| Continuity Only Model | COM | 0.74 | - | 0.147 | 2.563 | - | See Section 3.2.6 |
| Mixing Length Model | MLM | 0.83 | - | 6.923 | 15.692 | 15.255 | See Section 3.2.7 |
| Turbulent Viscosity Model | TVM | 2.38 | - | 0.636 | 9.394 | 9.52 | See Section 3.2.8 |
| Turbulent Viscosity and Turbulent Kinetic Energy Model | TVKEM | 1.63 | - | 0.47 | 5.616 | 6.186 | See Section 3.2.9 |

\* See Appendix E for calculation method.

### 3.1.2. Loss Function

A neural network loss function is the method that is used by the network to converge to the correct solution. The defining trait of a data-driven model is that the values predicted from the neural network (in this case, $\bar{u}$, $\bar{v}$, $\bar{p}$, $u'u'$, $u'v'$, and $v'v'$) are directly compared

against the DNS values from the dataset. The loss function for the data-driven model is a simple MAE (mean-averaged error). The error is calculated using,

$$Loss = MAE_{data} = \frac{1}{N} \sum_{i=1}^{N} |Y_i - \hat{Y}_p| \tag{9}$$

where $Y_i$ is the value from the dataset, and $\hat{Y}_p$ is the predicted value from the neural network. More information on neural networks for regression can be found in Appendix C.

3.1.3. Training

When using NNs for regression, it is common to normalise the input data between 0 and 1 as the nonlinearity introduced by the activation function is also between 0 and 1, and thus the model will converge more quickly when in this range [21]. While the architecture for the data-driven model uses ReLU, which scales between 0 and infinity, this would not account for any extreme values. The backpropagation calculates Euclidean distance, which cannot be computed accurately with features of different scales [22]. Therefore, the data will be normalised between 0 and 1 before being input into the neural network, using the range of each variable as the scale for that variable only.

The dataset procured contains the flow cases for 29 periodic hill geometries, 28 of which will be used for training and the final for testing. For a data-driven model to successfully extract nonlinear dynamics, it requires sufficient inputs to characterise the fluid flow individually, as discussed previously. It also requires a large training dataset to be able to represent the general flow case correctly. It is predicted that the simple MLP dense network with this dataset will capture main flow features and make predictions that are close but miss smaller scale or less common turbulence features due to this lack of data.

Overfitting is an issue in regression tasks when the neural network testing data validation does not improve or becomes worse relative to the training validation [23]. The model will implement early stopping, where a threshold of epochs with no improvement will cause the network to finish, reverting to the weights that give the best validation loss, to prevent overfitting the model.

The learning rate will start at $1 \times 10^{-3}$, and the threshold for the learning rate reduction is 12 epochs of no improvement to the validation loss. After these 12 epochs, the learning rate will be multiplied by 0.2, with a minimum bound of $1 \times 10^{-6}$. The threshold for the early stopping is 30 epochs of no validation loss improvement, where the best weights will be restored on end. If this condition is not met, the training will finish at 100 epochs. The batch size, or percentage of data fed into the network per epoch, is 128. The validation split for the neural network is 0.2, meaning 20% of the data are used to validate the updated weights, while 80% of the data are used to train the network to update the weights.

*3.2. Physics-Informed Neural Network*

3.2.1. Network Architecture

As with the data-driven model, the quantities to predict in the flow are the time-averaged streamwise velocity, $\bar{u}$, the time-averaged transverse velocity, $\bar{v}$, and the time-averaged pressure, $\bar{p}$. Additional requirements or compromises from these predictions will be made based on the implementation of the governing dynamics equations, seen in Sections 3.2.4–3.2.10. The network facilitates automatic backpropagation of nonlinear dynamics equations using the TensorFlow function tape.gradient, used to calculate tensor gradients such as the first-order partial differential terms in the governing flow equations.

The network architecture is adapted from a physics-informed network architecture proposed by Eivazi et al. [6]. This architecture consists of a similar dense MLP architecture as the data-driven model, but due to the nature of the loss function the convergence is less of a concern, so a stack of 8 layers with 20 hidden nodes each will be used for simplicity. The changes made to incorporate this implementation include the modification of the data loading to accommodate the dataset, and modification of the governing equations and

loss function, as well as changes to boundary condition enforcement and predictions to accommodate different governing equation sets.

### 3.2.2. Loss Function

The loss function for the PINN is different from the data-driven model. The loss function is a combination of calculated errors,

$$Loss = MSE_{bc} + MSE_{pde} \tag{10}$$

There $MSE_{bc}$ is a mean averaged error taken at the boundary conditions, and $MSE_{pde}$ is the mean squared error of the governing partial differential equations. The implementation of boundary condition enforcement is an MSE error, like the MAE formula used in the data-driven model. The formula for this MSE loss is given,

$$MSE_{bc} = \frac{1}{N} \sum_{i=1}^{N} |Y_i - \hat{Y}_p| \tag{11}$$

For any closed system to converge to a single solution, boundary conditions must be provided for each predicted variable. This does not necessarily mean that the equations cannot be solved without all variables' boundary conditions being provided, especially in the case of a neural network that could potentially compensate for inaccurate predictions with adjustment of the nodal weights based on flow features. This will be further explored in the different models.

As proposed by Eivazi et al., the loss function will be split into two training steps, 'supervised' and 'unsupervised' steps. The supervised step will involve the MSE$_{bc}$ and MSE$_{pde}$, for a small number of iterations to ensure that the flow fields converge to the correct flow state. When the boundary conditions are enforced, it is beneficial to remove the *MSEbc* enforcement condition to reduce the number of calculations per iteration, and thus reduce the solve time. After the initial training, the loss function will be reduced to only the *MSEpde* term, to converge the flow field to the correct solution in an unsupervised method. To monitor the impact of this, the inlet and outlet (used in this case as the boundary conditions) velocity profiles in the streamwise and transverse directions will be monitored, and if they are significantly different then the number of training iterations in the first step may be changed. It was found that 1000 epochs was generally sufficient to converge to the correct solution at the boundary, so this is used for all PINNs during testing.

### 3.2.3. Training

Like the data-driven model, the data will be normalised between 0 and 1 to improve convergence before being input to the neural network. Unlike the data-driven model, the only data-driven elements in this model are the data-enforced boundary conditions, with the 'test' data being the remaining nodes in the flow domain, governed by the governing equations. Like the data-driven model, an early stopping function is used to converge the model within a specified loss threshold, where, when no improvement is seen in a certain number of epochs, the model will stop training. This threshold is set to 50 epochs for the PINNs models. It was found that, to converge the models within the desired threshold, a learning rate scheduler was not required, so a constant learning rate of $1 \times 10^{-3}$ is used for all models.

### 3.2.4. Direct Reynolds Stress Model

The periodic hill flow conditions given in the dataset are governed by the 2D RANS equations as given in Equations (1)–(3). These have been derived in time-averaged form to match the flow field in the dataset, and separated into single partial differentials, which can easily be calculated using the tape.gradient function from TensorFlow, the Python library used to train the models. The equations are also rearranged to be equal to 0, which allows them to be directly summed to form the partial differential loss function,

$$MSE_{pde,1} = MSE_{continuity} + MSE_{momentum,x} + MSE_{momentum,y} \tag{12}$$

The MSE$_{\text{pde}}$ for all models will be similarly formed using the governing equations for each model in Table 6. Since the PINN will minimise the loss, the first method will involve a direct prediction of first-order stress terms, to directly close the momentum equations without needing to calculate these values using a numerical method or turbulence model. The full constraints of the model can be seen in Table 6.

### 3.2.5. Direct Reynolds Stress Model with Reduced Boundary Enforcement

Two more tests with direct Reynolds stress are performed, with alterations to be made to which variables are enforced as part of the boundary condition. The first will include $\bar{u}$ $\bar{v}$, and $\bar{p}$, but not first-order Reynolds stresses in boundary enforcement. The second will include only $\bar{u}$ and $\bar{v}$ in boundary enforcement. Since the neural network will minimise the loss function and can learn flow relationships, it is predicted that the neural network will maintain correct predictions on the variables that are enforced on the boundary conditions, and converge the variables that are not enforced on the boundary to a solution that follows,

$$\sum_{i=1}^{N}(V_{bi}) = \sum_{i=1}^{N}(V_{nbi}) \tag{13}$$

where $V_{bi}$ represents the variables enforced at the boundary and $V_{nbi}$ represents the variables not enforced at the boundary condition. This can also be interpreted that the sum of the predicted variables for a given point will be equal, regardless of whether the boundary condition is enforced. This will only be true based on the relationship between the variables and the equations in which they are present, in this case for first-order Reynolds stresses in the first test, and pressure and first-order Reynolds stresses in the second test.

It should be noted that, since the gradient forms of these variables are all that is present in the governing flow equations, the actual predicted values will not be enforced by Equation (13), but the gradients will, and this is what will be observed from the results to validate this hypothesis. This is also likely constrained based on the convergence of the neural network, as it is inherent with neural networks that they can converge to a solution that is not entirely the same as the ground truth, and the consistency of the convergence can vary based on the neural network, architecture, and convergence problem.

### 3.2.6. Continuity Only Model

It is proposed that the nature of a physics-informed neural network will converge a solution provided that the system is defined correctly, but this does not require the system to be numerically closed. In a direct numerical solver, continuity would not be sufficient by itself to close the system as it contains both $\bar{u}$ and $\bar{v}$, which both vary in space, thus requiring the additional momentum equations to solve numerically. However, assuming that predictions of time-averaged streamwise and transverse flows are sufficient for the application, it is proposed that the continuity will converge to the correct solution. The full model constraints can be seen in Table 6.

### 3.2.7. Mixing Length Model

As direct prediction is not possible with a traditional solver, various models have been developed to close the momentum equations. Most closure models use the Boussinesq assumption,

$$\tau_{ij} = \mu_t \left( 2S_{ij} - \frac{2}{3}\frac{\partial u_k}{\partial x_k}\delta_{ij} \right) - \frac{2}{3}\rho k \delta_{ij} \tag{14}$$

where $\mu_t$ is the turbulent viscosity, $S_{ij}$ is the stress tensor, $\delta_{ij}$ is the Kronecker delta, and $k$ is the turbulent kinetic energy. Derivations can be found in Appendix A.3. This equation

relates the first-order stresses to a derived turbulent viscosity term. This can be rewritten for single partial derivatives as is required by the TensorFlow tape.gradient function,

$$\overline{\tau_{ij}} = \mu_t \left( \left( \frac{\partial \overline{u_i}}{\partial x_j} + \frac{\partial \overline{u_j}}{\partial x_i} \right) \right) - \frac{2}{3} \rho k \delta_{ij} \tag{15}$$

A model that directly predicts turbulent viscosity can be used to close the Boussinesq assumption, which in turn can close the momentum and continuity equations. One method of approximating the turbulent viscosity is using the Prandtl mixing length model. This model, as used in [24], approximates the turbulent viscosity, $\mu_t$, using the following equation,

$$\mu_t = l_m^2 \sqrt{G} \tag{16}$$

where $G$ is the strain tensor, and $l_m$ is the mixing length,

$$G = 2 \left( \frac{\partial \overline{u}}{\partial x} \right)^2 + 2 \left( \frac{\partial \overline{v}}{\partial y} \right)^2 + \left( \frac{\partial \overline{u}}{\partial y} + \frac{\partial \overline{v}}{\partial x} \right)^2 \tag{17}$$

$$l_m = min(0.419d, 0.09d_{max}) \tag{18}$$

where $d$ is the distance from the wall. When approximating the first-order stress term, a turbulent kinetic energy, $k$, is also required. In the mixing length model, the turbulent kinetic energy is approximated using,

$$k = \frac{1}{2} \overline{u_i u_j} = \frac{1}{2} \left( \overline{u'u'} + \overline{v'v'} \right) \tag{19}$$

as outlined by [25], which allows the first-order stress terms to be defined,

$$\overline{\tau_{xx}} = 2\mu_t \left( \frac{\partial \overline{u}}{\partial x} + \frac{\partial \overline{u}}{\partial x} + \frac{\partial \overline{v}}{\partial y} \right) \tag{20}$$

$$\overline{\tau_{yy}} = 2\mu_t \left( \frac{\partial \overline{u}}{\partial x} + \frac{\partial \overline{v}}{\partial y} + \frac{\partial \overline{v}}{\partial y} \right) \tag{21}$$

$$\overline{\tau_{xy}} = \overline{\tau_{yx}} = \mu_t \left( \frac{\partial \overline{u}}{\partial x} + \frac{\partial \overline{v}}{\partial y} \right) \tag{22}$$

A full derivation can be found in Appendix A. Noting that some of the equations above can be simplified further, but have been left in the form shown to more easily link to the derivation presented. The full constraints of the model can be found in Table 6.

### 3.2.8. Turbulent Viscosity Model

It is known that the mixing length model is an oversimplification of the rich turbulence dynamics that occurs in fluid flow, specifically the relation of turbulent viscosity to a mixing length, implying that turbulent eddies maintain their identify over a certain distance, which is a simplification to relate mixing length to geometry. Since the physics-informed neural network will converge any included values to a solution that minimises the loss function, it is implied that the turbulent viscosity can be predicted directly by the neural network. This removes the reliance on the relation between geometry and mixing length and should improve the prediction accuracy. To test this, another model will be tested, but the turbulent viscosity will be directly predicted by the neural network. The same approximation for turbulent kinetic energy will be used as in Equation (19). The full constraints on the model can be seen in Table 6.

### 3.2.9. Turbulent Viscosity and Turbulent Kinetic Energy Model

The simplifications shown in Equation (19) can also be avoided by using the neural network to predict the values directly for turbulent kinetic energy as well. The first-order stress terms can then be directly predicted,

$$\overline{\tau_{xx}} = \mu_t \left( \frac{\partial \overline{u}}{\partial x} + \frac{\partial \overline{u}}{\partial x} \right) - \frac{2}{3} \rho k \tag{23}$$

$$\overline{\tau_{yy}} = \mu_t \left( \frac{\partial \overline{u}}{\partial y} + \frac{\partial \overline{v}}{\partial x} \right) \tag{24}$$

$$\overline{\tau_{xx}} = \mu_t \left( \frac{\partial \overline{v}}{\partial y} + \frac{\partial \overline{v}}{\partial y} \right) - \frac{2}{3} \rho k \tag{25}$$

The full constraints on the model can be seen in Table 6.

### 3.2.10. Computation

The CPU TensorFlow instruction set was compared timewise against the CUDA instruction set running through WSL (Windows Subsystems for Linux) on an Nvidia 3070 laptop GPU, as TensorFlow 2.14.0 does not support the CUDA instruction set on Windows (NVIDIA, Santa Clara, CA, USA). The overhead from WSL leads to poor performance and issues with RAM limitations loading the dataset into the neural network, so the CPU instruction set is used for all runs on native Windows 11.

### 3.3. Summary of PINNs Methods

Figure 2 and Table 4 provide a summary of the PINNs models described, and the conditions under which these models have been systematically evaluated.

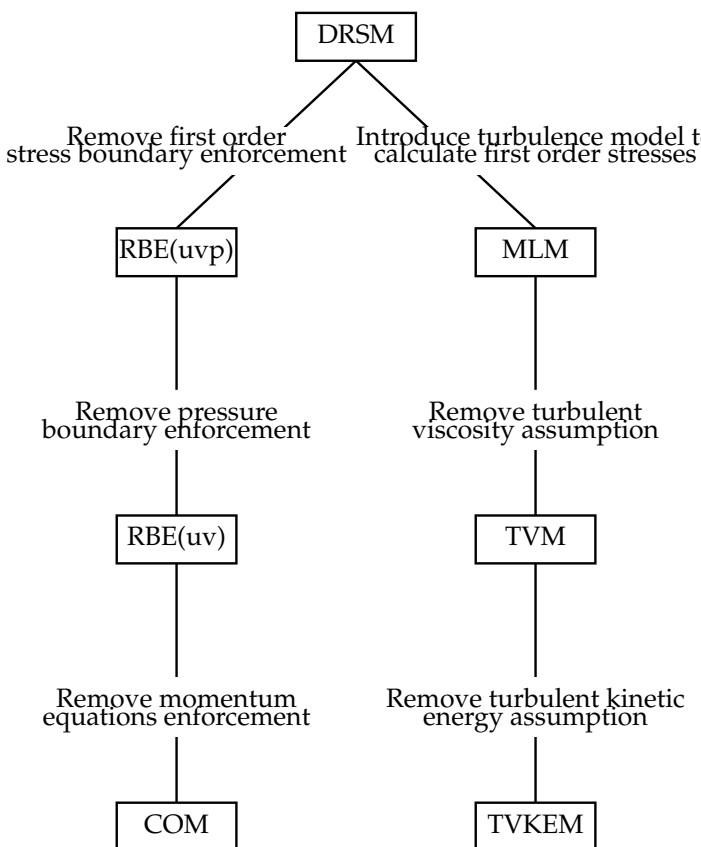

**Figure 2.** Methods summary. First approach to reduce number of governing equations by taking advantage of the ability to solve a numerically open system (**right**). Second approach by introducing turbulence modelling assumptions to simplify predictions (**left**).

## 4. Results

### 4.1. Data-Driven Model

The data-driven model's time-averaged streamwise velocity prediction in Figure 3 generally reflects the features of the DNS data. However, it fails to capture small-scale

turbulence features including boundary conditions at the inlet and backflow conditions at the trailing slope, where the model has not converged. Overall, the domain error is low at 2.85%.

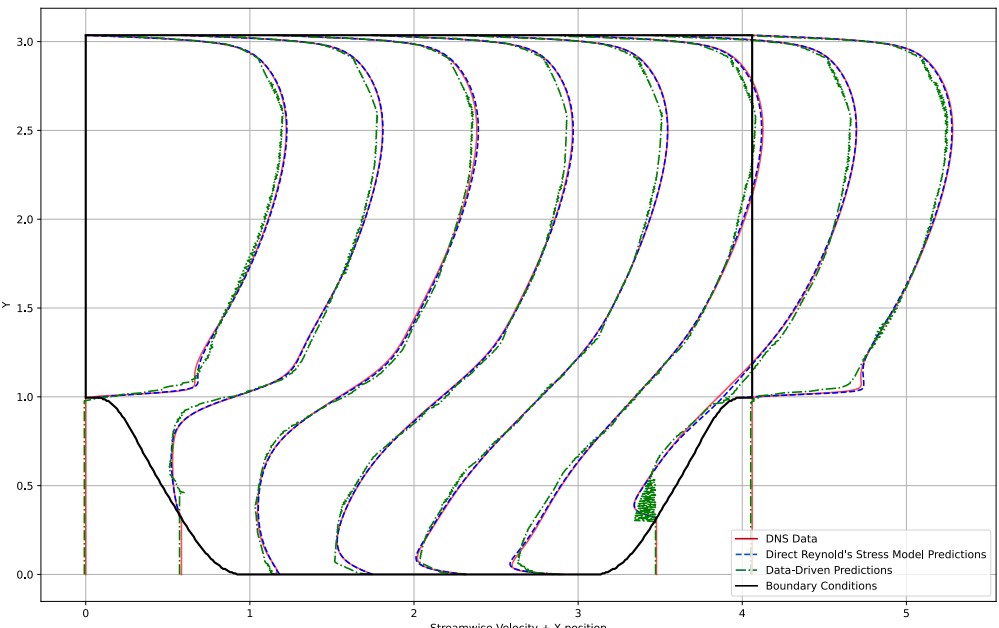

**Figure 3.** Time-averaged streamwise velocity, $\overline{u}$, data driven vs. DRSM vs. DNS.

The transverse velocity prediction in Figure 4 exhibits a convergence towards correct dynamics in bulk flow for the transverse velocity prediction by the data-driven model. It captures features such as upflow at the leading edge but with poor accuracy and is missing an upflow condition at the trailing slope. The domain error is much higher than the streamwise velocity at 67.5%.

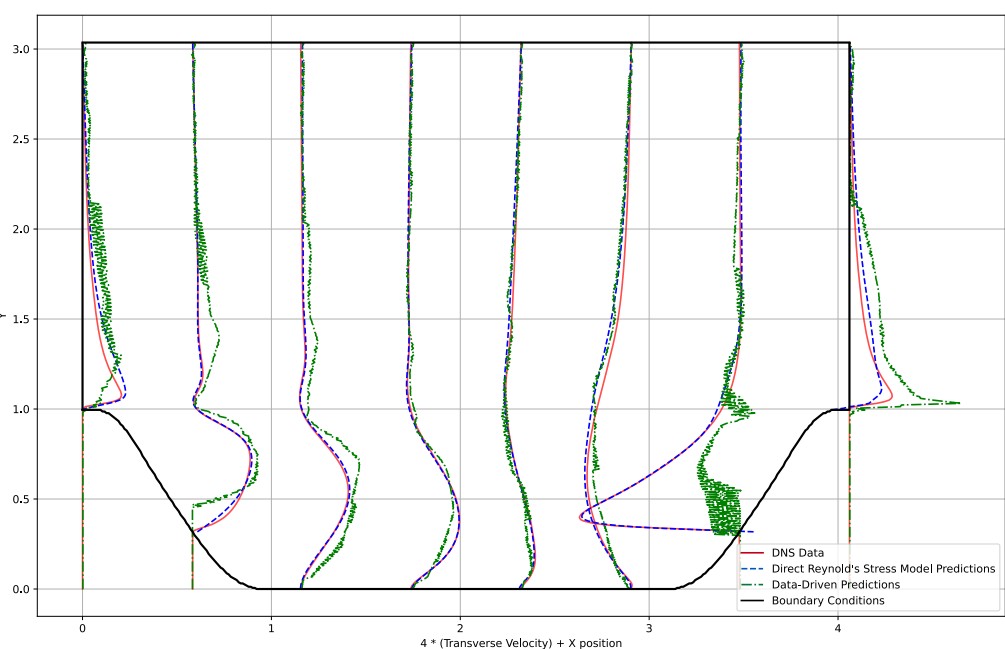

**Figure 4.** Time-averaged transverse velocity, $\overline{v}$, data driven vs. DRSM vs. DNS.

While there is an approximate shape of the pressure gradient shown in Figure 5, predictions are noisy and lack accuracy. The magnitude is overpredicted for negative

pressure in the centre of bulk flow, and positive pressure near the trailing edge. Effects on boundaries are also missed with very noisy predictions. The domain error is very high at 215.9%.

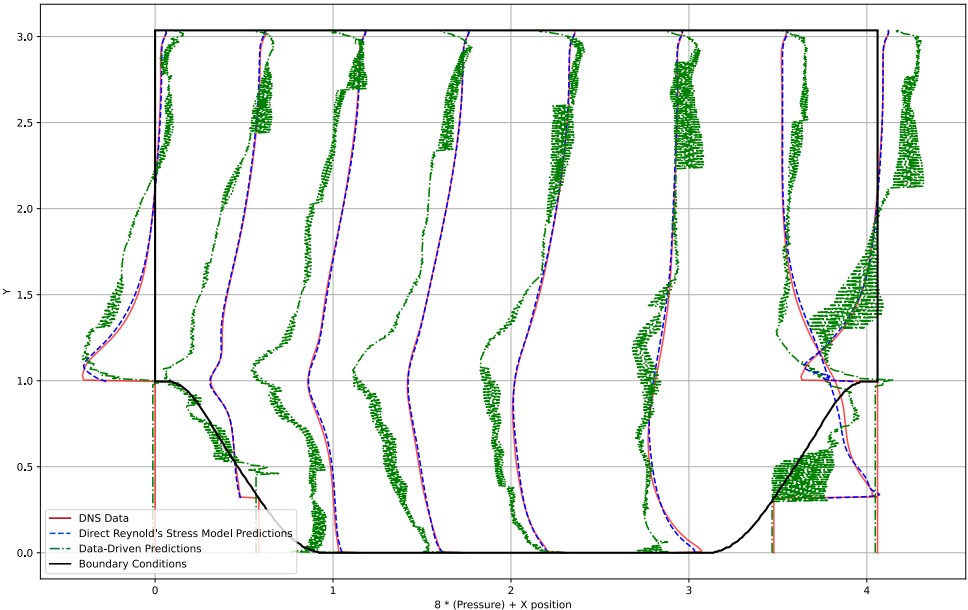

**Figure 5.** Time-averaged pressure, $\overline{p}$, data driven vs. DRSM vs. DNS.

### 4.2. PINNs Models

It is seen in Figure 3 that the DRSM can capture the flow much more accurately than the data-driven model. Elements that are not predicted well by the data-driven model, including the boundary condition at the inlet and outlet and the boundary at the trailing slope, are captured much more accurately, with a lower total error in the streamwise flow of 0.613%. This is evidenced visually through the streamwise velocity for PINNs models (Figure 6).

The time-averaged transverse velocity prediction of the DRSM PINN is compared against the DNS and the data-driven models in Figure 4. The DRSM predictions more accurately capture flow features including the boundary feature on the inlet and outlet, as well as central flow regions, as shown in Figure 7, Section 2. The overall domain error is lower with the DRSM, with only a 9.24% error. It is noted that there is significantly more error in the transverse velocity predictions than in the streamwise predictions.

Figure 5 shows that the DRSM PINN can predict the averaged pressure field much more accurately than the data-driven model, with a significantly lower error of 4.78%. The predictions are much more accurate in the bulk flow, and significantly improved in the boundaries and on the leading and trailing edges; however, there are still differences at the boundary of the trailing edge and the outlet condition.

The variables that were removed from the boundary conditions in the $RBM_{uvp,uv}$ are not correctly predicted, as can be seen in the pressure field error of 253.68% for $RBM_{uv}$. However, it is interesting to note that, by reducing the enforced variables on the boundary, the predictions for the variables that were enforced improve on average, with the error for streamwise velocity reducing from 0.613% to 0.309% when not enforcing first-order Reynolds stress terms, with a further drop to 0.233% when not enforcing pressure $\rho$ either. A similar trend is seen for transverse velocity. The effect of reducing the boundary conditions on the solve time is not clear, with a reduction and increase (Table 6).

As shown in Figures 6 and 7, the COM produces the most accurate results for streamwise and transverse velocities, with overall domain errors of 0.147% and 2.562%, respectively. This includes the most accurate capturing of flow features such as the boundary flow at the leading and trailing slopes. The solve time is also comparatively low at only 0.74 h.

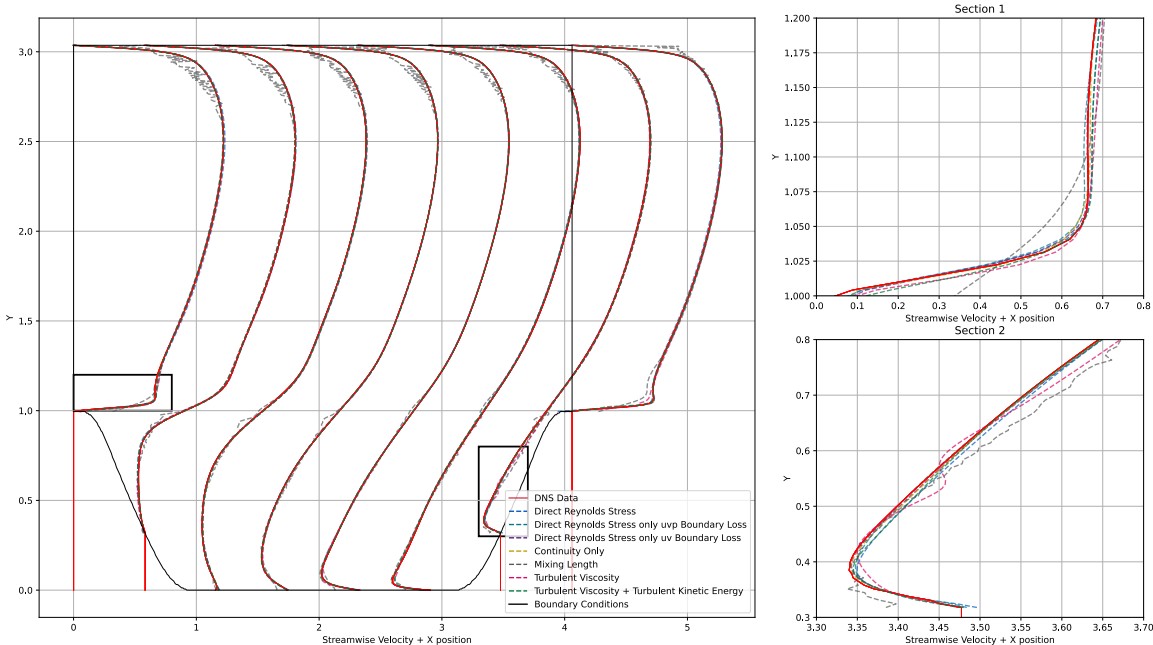

**Figure 6.** Time-averaged streamwise velocity, $\overline{u}$, PINNs compared.

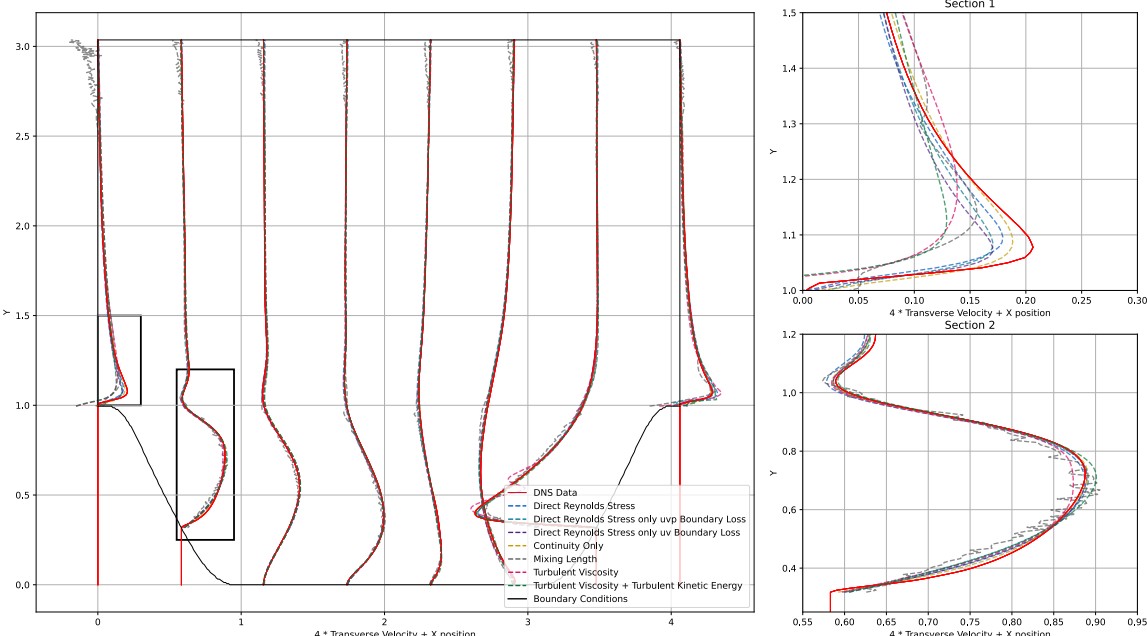

**Figure 7.** Time-averaged transverse velocity, $\overline{v}$, PINNs compared.

As shown in Figures 6–8, the MLM does not converge to an accurate solution near the boundary conditions for the streamwise and transverse velocity fields, or the pressure field, with the highest PINNs method error for streamwise velocity at 6.92%, and highest enforced error for transverse velocity and pressure at 15.7% and 15.3%, respectively. The predictions capture the general flow but are particularly poor for streamwise velocity near the boundaries, especially the top boundary. The solve time however for this model is very low, at only 0.74 h.

As shown in Figures 6–8, the TVM has a much higher accuracy than the MLM, with the overall streamwise error reduced to 0.64%, transverse velocity error reduced to 9.29%, and pressure field error reduced to 9.52%. The poor convergence at the boundaries seen in

the MLM is not present, with much better convergence at the leading and trailing slopes. The solve time is much higher than the MLM, at 2.38 h.

The accuracy is further improved with the TVKEM, shown in Figures 6–8. The overall error is further decreased from the TVM, with the streamwise velocity error reduced to 0.47%, transverse velocity error reduced to 5.62%, and pressure field error reduced to 6.19%. The solve time is still higher than the MLM at 1.63 h but is much lower than the TVM.

Figures 9 and 10 show the gradient of the first-order stress terms in the momentum in x and momentum in y equations, respectively. It is shown that the bulk prediction for the gradients is similar between all models and the DNS, with discrepancies at the boundaries where large positive and negative gradients are present.

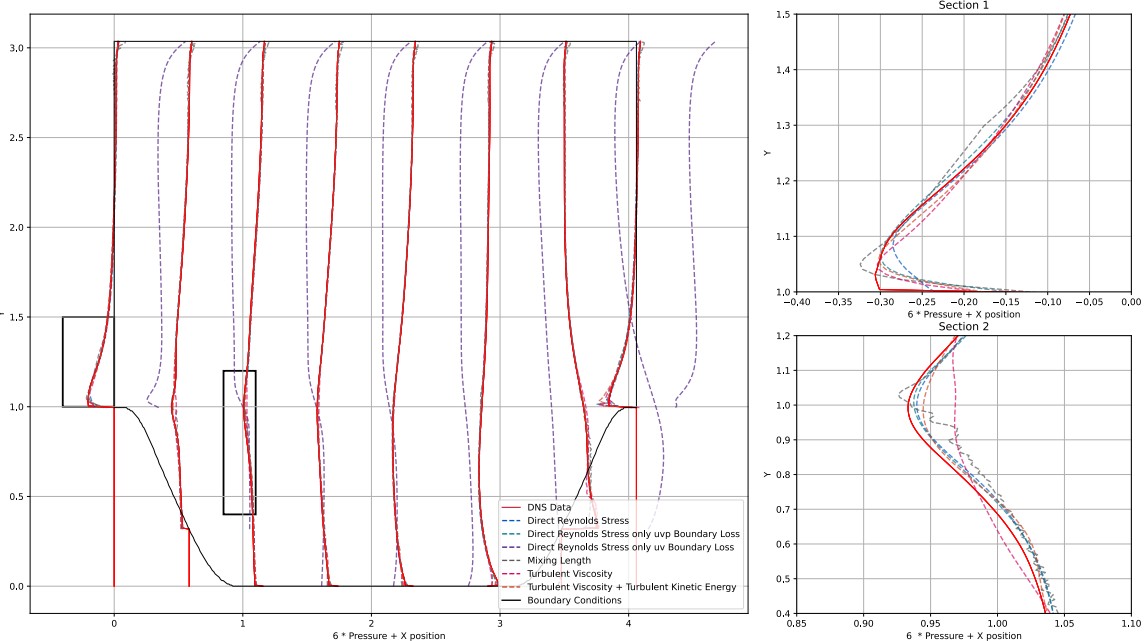

**Figure 8.** Time-averaged pressure, $\overline{p}$, PINNs compared.

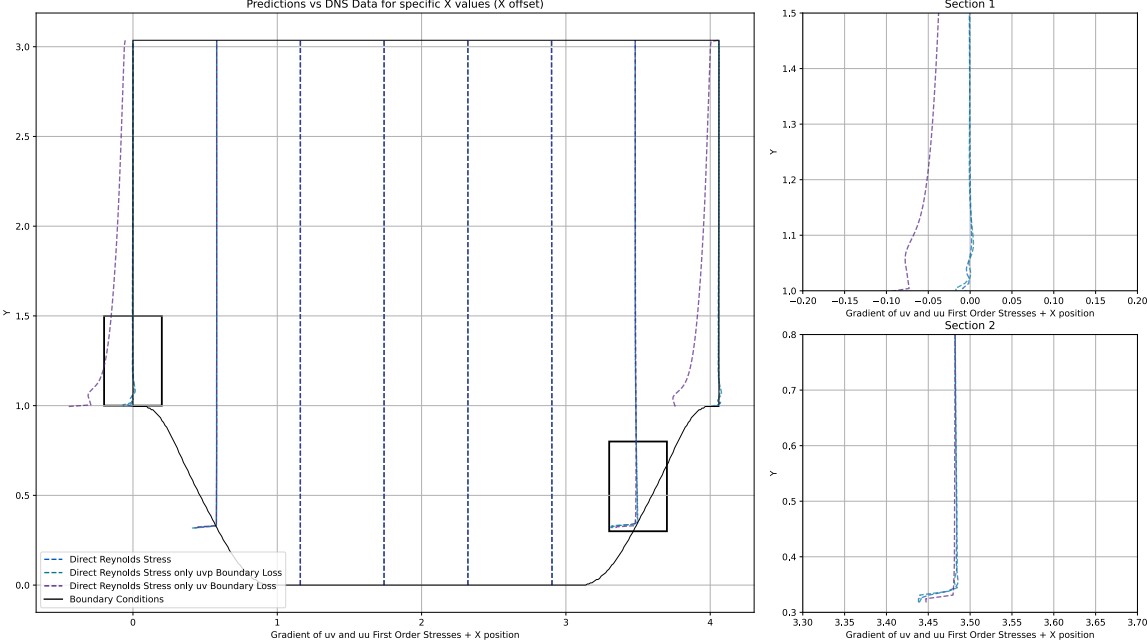

**Figure 9.** Gradient of $u'u'$ and $u'v'$ first-order stresses from $x$ momentum equation for RBM models.

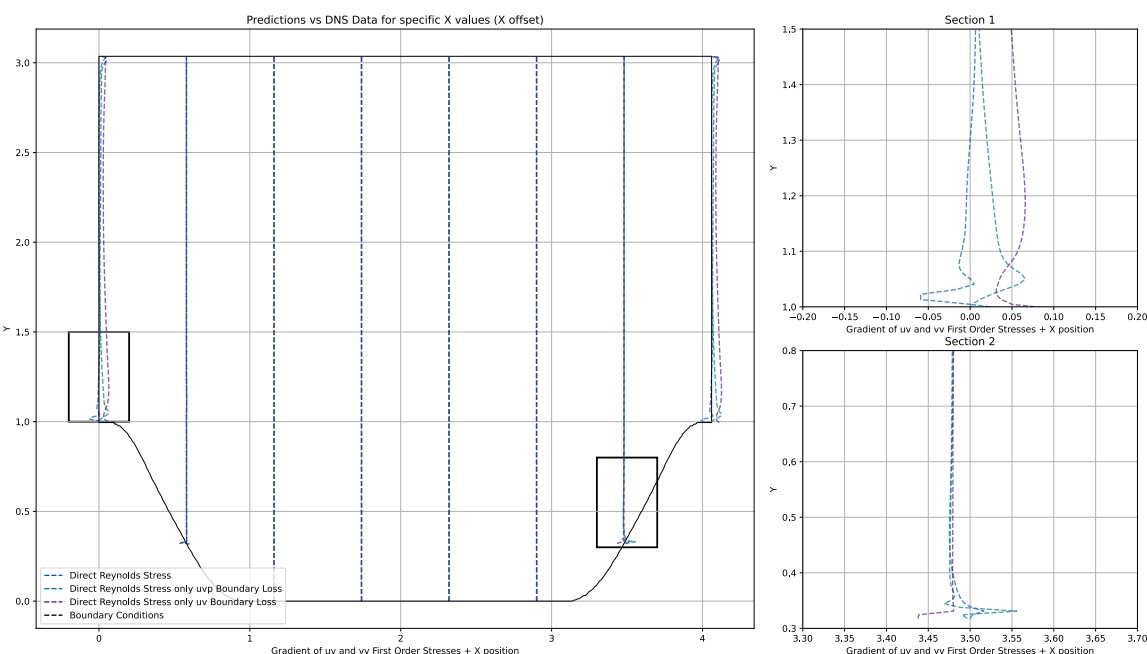

**Figure 10.** Gradient of $u'u'$ and $u'v'$ first-order stresses from $y$ momentum equation for RBM models.

## 5. Discussion

### 5.1. Data-Driven Model

The discrepancy between the data-driven model predictions and DNS is likely due to the limited dataset. The small size of the dataset does not allow the neural network to capture the complex nonlinear dynamics between geometric inputs and flow outputs for a general case. Slices of streamwise velocity data (Appendix G) indicate similar profiles among different geometry flow cases for the inlet, supporting the hypothesis that similarity in the data leads to more accurate characterisation of flow and thus more accurate prediction. Discrepancies at the boundary conditions at the trailing edge are likely due to poor normalisation of data after concatenation into an array containing all datasets, leading to differences in position when normalised, which affect predictions, especially at domain edges.

The normalisation limitation of the data-driven model is clearer in the transverse velocity prediction, where the backflow condition at the trailing slope is entirely missed by the data-driven model. This flow feature is only present in three of the 29 flow cases (Appendix G) at this normalised x value, which suggests that this prediction failure is a result of the sparse data problem, where features that are sparse in the dataset are not captured correctly because of the equal weighting of all features. This could be potentially resolved by modification of the dataset to ensure that the domain lengths are similar after normalisation. In this case, the backflow condition on the trailing slope would be constant in position. This would aid the neural network in identification of this feature but would still leave the inaccuracy issue seen in the rest of the flow features, e.g., on the inlet flow condition.

The noisy and inaccurate predictions of the pressure field again suggest that the dataset is not sufficient to capture the dynamics required to predict the pressure field. The pressure field is significantly different in all flow cases, with similar gradients only between a few of the flow cases (Appendix G). This supports the theory that the data in this case are too different to provide accurate predictions, and more flow cases would be needed to construct an accurate pressure field.

Again, it should be noted that the methods for this data-driven model are simple as a based point of comparison; however, there are data-driven frameworks proposed in literature that can make more accurate predictions. Volpiani et al. proposed a similar data-driven architecture [26], with a larger set of input features, to allow the neural network to

adapt to relevant features. This included turbulent flow features such as strain rate, rotation rate, turbulent intensity, etc., and these were separately normalised with customised factors to avoid moving turbulent features. While no error or accuracy metrics are provided, it can be seen from the graphical results that the additional neural network inputs do improve predictions, with both models tested capturing more streamwise velocity detail at the boundary conditions, particularly the bottom wall. The transverse predictions are also improved, more accurately predicting the downflow conditions at the leading slope as the domain is normalised to preserve the turbulence model's position. However, though the predictions are improved, all models show some error compared with the proposed PINNs approaches.

*5.2. PINNs Models*

Comparing the error between the DRSM and RBM, the RBMs generally have a lower error in the enforced variables than the DRSM (Table 6). This is a result of the model being constrained by fewer factors, and fewer constraints allows the model to converge to a solution more quickly, as there are fewer variables that are being enforced. Since the relationship between the variables in the governing equations is based on the gradient of the variables (with respect to the streamwise or transverse direction), there is nothing to enforce the singular correct solution in the base values, which explains the high overall domain error. The results from Figures 9 and 10 prove this, as the gradient values that are part of the governing equations for the bulk flow are very similar. There is still a small deviation between the models, which is likely due to the limited iterations for the PINN to converge, so the exact solution is not reached. It is hypothesised that a run that is allowed to continue training to a lower convergence threshold would see the DRSM coming closer and eventually surpassing the RBMs; however, in an engineering application, time is an important consideration. Since there is no clear benefit to reducing the boundary enforcement in terms of solve time, it is hypothesised that the reduction in time is dependent on the initial weights in the neural network from the first training step, or the 'first guess' at these variables. If this value is reasonably close, then the time is reduced, whereas, if the value is not close, the time is increased. Further testing is required to understand this effect, and enforcing a single or very few nodes of data and comparing the time vs full boundary enforcement for the variable would prove this hypothesis. Should this hypothesis be true, the most efficient training method would be to include the full set of boundaries at the initial 1000 epoch training, but remove them after to reduce the constraints on the model while keeping the initial values close to the correct values.

There is a significant improvement in the implementation of this neural network from [6], which produced higher errors of 2.8%, 19.7%, and 8.6% in streamwise velocity, transverse velocity, and pressure, respectively, for a periodic hill test case. This is not directly comparable, as the dataset and Reynolds number are different, and the improvement shown is likely due to a different convergence threshold; however, it is useful to note that the PINN can produce lower errors, given optimisation of the solver for the problem.

It is noted that the error in the predicted variables for the COM is the lowest of any PINNs model. Since this model contains the fewest boundaries and equations to satisfy, the solution is reached much more quickly, with each epoch taking less time as fewer gradients and equations must be calculated. An obvious limitation of this model is that, by limiting the equations, the predicted variables are also limited, as there are no predictions made for the pressure field. Since any equations that govern the flow must be correct to converge to a correct solution, and have boundary conditions provided, if these pressure predictions are required then the momentum equations must be added to the loss function. However, if only the velocity fields are needed, a continuity only model is shown to be effective with low error and solve time. This is particularly interesting, as effects of pressure and first-order stresses are being ignored in the loss function but appear to still affect the solution, likely a result of the boundary enforcement technique used during training. Further work is required to understand the difference in the nodal weight adjustment between a COM and

DRSM, or further proof is required on alternative flow cases, particularly cases with heavy turbulence characteristics such as vortex shedding, flow layer separation, and backflow, such as a backwards-facing step.

The lack of accuracy of the MLM at the boundary conditions can be explained by the model and its assumptions. Prandtl's mixing length model is known to have several limitations, including the lack of a physical basis for the assumed proportionality between the turbulent viscosity and the velocity gradients that make up the strain tensor [27,28]. The model includes an overly simplistic assumption of turbulent viscosity, which has a constant mixing length for the flow case. This is likely the largest contributor to the poor convergence near the boundaries, as the periodic hill has changing geometries on the bottom boundary. This results in the value of the mixing length calculated not being applicable across the whole domain due to the changes in geometry.

Removing this mixing length assumption and the modelling of the turbulent viscosity without relying on the strain tensor by directly predicting the values resolves the convergence issue at the boundary conditions. This indicates that this assumption is the limiting assumption in the MLM between the mixing length assumption and the turbulent kinetic energy assumption. However, the model still contains the turbulent kinetic energy assumption, which is also a simplification. Removing both assumptions leaves a model that is not numerically closed but relies only on the Boussinesq assumption. The improvement of the TVKEM from the TVM implies that the TVKEM is significantly impacting the accuracy of the model for this flow case. It is noted that using the Boussinesq assumption itself does not impact the accuracy of the model; however, the poor assumptions that are used to calculate the values required in the Boussinesq assumption should be avoided, where possible, as they impact the solver accuracy.

In our current study, there is a novel proposition of exploiting numerically derived unclosed equation sets to aid in efficient solving of problems related to turbulence. Specifically where the equation sets used to constrain PINNs are based on a numerically derived reduced order model. Existing studies which employ a reduced order model, for turbulence, can do so effectively but may require understanding of the geometry or an actual insight into the data itself under given conditions [9]. It has been shown that neural networks can reduce the dimensionality of a governing equation [11], and that they can be applied successfully to parametric systems [10]. The results obtained in this current study are broadly in agreement with literature, such as Pioch et al., 2023 [7] which indicate that physics-informed neural networks are a significant improvement in solve time compared to a data driven approach. However, so far turbulence problems have not evaluated unclosed problems, where more variables are solved than equations sets available. This approach, as used in our current study has been particularly effective when data-driven and PINNs have been combined, leading to a time-efficient and high-accuracy solution. This approach, while evaluated for a turbulence benchmark model, has wider applications where complex complex flow is involved, such as in multi-phase and multi-scale flow models which when solved using a traditional numerical modelling approach have long solution times.

## 6. Conclusions

This study compared a data-driven turbulence modelling approach to different configurations of a PINNs architecture to predict flow fields of a period hill of Reynolds flow 5600, governed by 2D incompressible RANS equations. It was shown that the simple data-driven model is accurate to 2.9% and 67.5% error in streamwise and transverse velocities, respectively, and that this approach is constricted by the small number of flow cases and sparse data of velocity features between the flow cases. The numerically closed DRSM PINN was shown to have a significantly reduced error (1.2% and 6.4%, respectively), with further improvement using reduced-boundary models and reduced-equation models, with the COM having a dramatically reduced solve time and error in fluid velocity predictions compared with all other approaches. It was also shown that introducing poor assumptions in the governing equations affects the convergence of the solver, with the MLM model

showing significant error at the boundary conditions. The evaluation of a turbulence benchmark has enabled the effectiveness of the PINNs to be assessed. Critically, this study has evaluated a novel approach by which to implement a reduced order model. This approach is time-efficient for solving turbulence problems in cases of sparse-data using an unclosed approach to its solution. In conclusion, a numerically derived unclosed equation set can be efficiently implemented with neural networks to reliably model turbulent systems.

**Author Contributions:** W.F.: conceptualization, data curation, formal analysis, methodology, software, investigation, validation, writing—original draft and editing. B.S.: analysis, writing—review and editing. J.C.: conceptualization, analysis, writing—review and editing. M.C.: conceptualization, analysis, methodology, writing—review and editing. D.M.E.: conceptualization, analysis, supervision, project administration, methodology, writing—review and editing. All authors have read and agreed to the published version of the manuscript.

**Funding:** J.C. was supported by the Engineering and Physical Sciences Research Council [grant number: **EP/T517926/1**].

**Data Availability Statement:** Data and relevant code for this research work are stored in GitHub [29].

**Conflicts of Interest:** The authors declare no conflicts of interest.

## Abbreviations

The following abbreviations are used in this manuscript:

| | |
|---|---|
| PINNs | Physics-informed neural network |
| CFD | Computational fluid dynamics |
| DNS | Direct numerical simulation |
| ROM | Reduced-order modelling |
| RANS | Reynold's averaged Navier–Stokes |
| MLP | Multilayer perceptron |
| MAE | Mean averaged error |
| MSE | Mean squared error |
| DRSM | Direct Reynold's stress model |
| $RBM_{uvp}$ | Direct Reynold's stress model with reduced boundary enforcement of only $u$, $v$, and $p$ |
| $RBM_{uv}$ | Direct Reynold's stress model with reduced boundary enforcement of only $u$ and $v$ |
| COM | Continuity only model |
| MLM | Mixing length model |
| TVM | **Turbulent viscosity** model |
| TVKEM | Turbulent viscosity and turbulent kinetic energy model |
| KOM | K-Omega model |
| $\bar{x}$ | Bar indicates time-averaged component |
| $x'$ | Prime indicates fluctuating component |
| $\bar{u}$ | Time-averaged streamwise velocity |
| $\bar{v}$ | Time-averaged transverse velocity |
| $\rho$ | Density (kg/m$^3$) |
| $\mu$ | Kinematic viscosity (m$^2$/s) |
| $\tau_{ij}$ | First-order Reynold's stress wrt. i and j |
| $\alpha$ | Geometric slope parameter |
| $h$ | Domain height (m) |
| $l$ | Domain length (m) |
| $Y_i$ | Dataset value |
| $Y_p$ | Neural network predicted value |
| $V_{bi}$ | Boundary-enforced variables |
| $V_{nbi}$ | Non-boundary-enforced variables |
| $\bar{p}$ | Time-averaged pressure |
| $\mu$ | Turbulent viscosity (kg/ms) |
| $S_{ij}$ | Stress tensor |
| $\delta_{ij}$ | Kronecker delta |
| $k$ | Turbulent kinetic energy |

| | |
|---|---|
| $d$ | Distance from the wall (m) |
| $G$ | Strain tensor |
| $l_m$ | Mixing length |

## Appendix A

*Appendix A.1. 2D Stationary Equations—Continuity*

The 2D continuity equation for incompressible flow is given by,

$$\frac{\partial u}{\partial x} + \frac{\partial v}{\partial y} = 0 \tag{A1}$$

The Reynolds decomposition can be used to separate the velocities into their mean and fluctuating components,

$$u = \bar{v} + v' \tag{A2}$$

$$u = \bar{v} + v' \tag{A3}$$

where $\bar{u}$ and $\bar{v}$ are the mean velocity components, and the prime denotes fluctuating components.

$$\frac{\partial(\bar{u} + u')}{\partial x} + \frac{\partial(\bar{v} + v')}{\partial y} = 0 \tag{A4}$$

Applying time averaging over time *a*,

$$\frac{1}{t_a} \int_0^{t_a} \left[ \frac{\partial(\bar{u} + u')}{\partial x} + \frac{\partial(\bar{v} + v')}{\partial y} \right] dt = 0 \tag{A5}$$

$$\frac{1}{t_a} \int_0^{t_a} \left( \frac{\partial \bar{u}}{\partial x} + \frac{\partial u'}{\partial x} + \frac{\partial \bar{v}}{\partial y} + \frac{\partial v'}{\partial y} \right) dt = 0 \tag{A6}$$

The Reynolds stress decomposition implies that, for stationary flow,

$$\frac{1}{t_a} \int_0^{t_a} \frac{\partial u'}{\partial x} dt = \frac{1}{t_a} \int_0^{t_a} \frac{\partial v'}{\partial y} dt = 0 \tag{A7}$$

Therefore,

$$\frac{1}{t_a} \int_0^{t_a} \frac{\partial \bar{u}}{\partial x} dt + \frac{1}{t_a} \int_0^{t_a} \frac{\partial \bar{v}}{\partial y} dt = 0 \tag{A8}$$

Which simplifies to,

$$\frac{\partial \bar{u}}{\partial x} + \frac{\partial \bar{v}}{\partial y} = 0 \tag{A9}$$

*Appendix A.2. 2D Stationary Equations—Momentum in x and y*

The 2D momentum equation is given by,

$$\rho \left( \frac{\partial u}{\partial t} + u \frac{\partial u}{\partial x} + v \frac{\partial u}{\partial y} \right) = -\frac{\partial p}{\partial x} + \mu \left( \frac{\partial^2 u}{\partial x^2} + \frac{\partial^2 u}{\partial y^2} \right) \tag{A10}$$

By dividing by $\rho$ on both sides,

$$\left( \frac{\partial u}{\partial t} + u \frac{\partial u}{\partial x} + v \frac{\partial u}{\partial y} \right) = -\left( \frac{1}{\rho} \right) \frac{\partial p}{\partial x} + \nu \left( \frac{\partial^2 u}{\partial x^2} + \frac{\partial^2 u}{\partial y^2} \right) \tag{A11}$$

The Reynolds stress composition for velocity and pressure is denoted,

$$u = \bar{u} + u' \tag{A12}$$

$$p = \bar{p} + p' \tag{A13}$$

where $\bar{u}$ and $\bar{p}$ are the mean streamwise velocity and pressure components, respectively, and the prime denotes fluctuating components. Substituting this into the momentum equation,

$$\left( \frac{\partial(\bar{u}+u')}{\partial t} + (\bar{u}+u')\frac{\partial(\bar{u}+u')}{\partial x} + (\bar{v}+v')\frac{\partial(\bar{u}+u')}{\partial y} \right) \tag{A14}$$

$$= -\left(\frac{1}{\rho}\right)\frac{\partial(\bar{p}+p')}{\partial x} + \nu\left( \frac{\partial^2(\bar{u}+u')}{\partial x^2} + \frac{\partial^2(\bar{u}+u')}{\partial y^2} \right)$$

Time averaging both sides,

$$\frac{1}{t_a}\int_0^{t_a}\left[ \frac{\partial(\bar{u}+u')}{\partial t} + (\bar{u}+u')\frac{\partial(\bar{u}+u')}{\partial x} + (\bar{v}+v')\frac{\partial(\bar{u}+u')}{\partial y} \right]dt \tag{A15}$$

$$= \frac{1}{t_a}\int_0^{t_a}\left[ -\left(\frac{1}{\rho}\right)\frac{\partial(\bar{p}+p')}{\partial x} + \nu\left( \frac{\partial^2(\bar{u}+u')}{\partial x^2} + \frac{\partial^2(\bar{u}+u')}{\partial y^2} \right) \right]dt$$

This can be decomposed to,

$$\frac{1}{t_a}\int_0^{t_a}\frac{\partial\bar{u}}{\partial t}dt + \frac{1}{t_a}\int_0^{t_a}\frac{\partial u'}{\partial t}dt \tag{A16}$$

$$+\left( \frac{1}{t_a}\int_0^{t_a}\bar{u}\frac{\partial\bar{u}}{\partial x}dt + \frac{1}{t_a}\int_0^{t_a}\bar{u}\frac{\partial u'}{\partial x}dt + \frac{1}{t_a}\int_0^{t_a}u'\frac{\partial\bar{u}}{\partial x}dt + \frac{1}{t_a}\int_0^{t_a}u'\frac{\partial u'}{\partial x}dt \right)$$

$$+\left( \frac{1}{t_a}\int_0^{t_a}\bar{v}\frac{\partial\bar{u}}{\partial y}dt + \frac{1}{t_a}\int_0^{t_a}\bar{v}\frac{\partial u'}{\partial y}dt + \frac{1}{t_a}\int_0^{t_a}v'\frac{\partial\bar{u}}{\partial y}dt + \frac{1}{t_a}\int_0^{t_a}v'\frac{\partial u'}{\partial y}dt \right)$$

$$= \frac{1}{t_a}\int_0^{t_a}\left(-\frac{1}{\rho}\right)\frac{\partial\bar{p}}{\partial x}dt + \frac{1}{t_a}\int_0^{t_a}\left(-\frac{1}{\rho}\right)\frac{\partial p'}{\partial x}dt + \frac{\nu}{t_a}\int_0^{t_a}\left(-\frac{1}{\rho}\right)\frac{\partial^2\bar{u}}{\partial x^2}dt$$

$$+\frac{\nu}{t_a}\int_0^{t_a}\left(-\frac{1}{\rho}\right)\frac{\partial^2 u'}{\partial x^2}dt + \frac{\nu}{t_a}\int_0^{t_a}\left(-\frac{1}{\rho}\right)\frac{\partial^2\bar{u}}{\partial y^2}dt + \frac{\nu}{t_a}\int_0^{t_a}\left(-\frac{1}{\rho}\right)\frac{\partial^2 u'}{\partial y^2}dt$$

Time averages of fluctuating components as implied by the Reynolds stress decomposition,

$$\frac{1}{t_a}\int_0^{t_a}\frac{\partial u'}{\partial t}dt = \frac{\bar{u}}{t_a}\int_0^{t_a}\frac{\partial u'}{\partial x}dt = \frac{\bar{v}}{t_a}\int_0^{t_a}\frac{\partial u'}{\partial y}dt = \frac{1}{t_a}\int_0^{t_a}\left(-\frac{1}{\rho}\right)\frac{\partial p'}{\partial x}dt = \frac{\nu}{t_a}\int_0^{t_a}\left(-\frac{1}{\rho}\right)\frac{\partial^2 u'}{\partial x^2}dt$$

$$= \frac{\nu}{t_a}\int_0^{t_a}\left(-\frac{1}{\rho}\right)\frac{\partial^2 u'}{\partial y^2}dt = 0 \tag{A17}$$

However, it is important to note that, while the time average of velocity and pressure fluctuation is zero, the variance is not zero. In this case,

$$\frac{1}{t_a}\int_0^{t_a}u'\frac{\partial u'}{\partial x}dt \neq 0, \frac{1}{t_a}\int_0^{t_a}u'\frac{\partial u'}{\partial x}dt \neq 0 \tag{A18}$$

The average velocity does not change with time,

$$\frac{\partial\bar{u}}{\partial t} = 0 \tag{A19}$$

Which leaves,

$$\left( \frac{1}{t_a}\int_0^{t_a}\bar{u}\frac{\partial\bar{u}}{\partial x}dt + \frac{1}{t_a}\int_0^{t_a}u'\frac{\partial u'}{\partial x}dt \right) + \left( \frac{1}{t_a}\int_0^{t_a}\bar{v}\frac{\partial\bar{u}}{\partial y}dt + \frac{1}{t_a}\int_0^{t_a}v'\frac{\partial u'}{\partial y}dt \right) \tag{A20}$$

$$= \frac{1}{t_a}\int_0^{t_a}\left(-\frac{1}{\rho}\right)\frac{\partial\bar{p}}{\partial x}dt + \frac{\nu}{t_a}\int_0^{t_a}\left(-\frac{1}{\rho}\right)\frac{\partial^2\bar{u}}{\partial x^2}dt + \frac{\nu}{t_a}\int_0^{t_a}\left(-\frac{1}{\rho}\right)\frac{\partial^2\bar{u}}{\partial y^2}dt$$

This can be simplified,

$$\overline{u}\frac{\partial \overline{u}}{\partial x} + \overline{v}\frac{\partial \overline{u}}{\partial y} + \overline{u'\frac{\partial \overline{u}'}{\partial y}} + \overline{u'\frac{\partial \overline{u}}{\partial x}} = \left(-\frac{1}{\rho}\right)\frac{\partial \overline{p}}{\partial x} + v\frac{\partial^2 \overline{u}}{\partial x^2} + v\frac{\partial^2 \overline{u}}{\partial y^2} \tag{A21}$$

This can be rearranged,

$$\overline{u}\frac{\partial \overline{u}}{\partial x} + \overline{v}\frac{\partial \overline{u}}{\partial y} - \left(-\frac{1}{\rho}\right)\frac{\partial \overline{p}}{\partial x} - v\left(\frac{\partial^2 \overline{u}}{\partial x^2} + \frac{\partial^2 \overline{u}}{\partial y^2}\right) + \overline{v'\frac{\partial \overline{u}'}{\partial y}} + \overline{u'\frac{\partial \overline{u}'}{\partial x}} = 0 \tag{A22}$$

where the final two terms represent the first-order Reynolds stresses. The same can be performed for the momentum equation in y,

$$\rho\left(\frac{\partial v}{\partial t} + u\frac{\partial v}{\partial x} + v\frac{\partial v}{\partial y}\right) = -\frac{\partial p}{\partial y} + \mu\left(\frac{\partial^2 v}{\partial x^2} + \frac{\partial^2 v}{\partial y^2}\right) \tag{A23}$$

Which gives,

$$\overline{u}\frac{\partial \overline{v}}{\partial x} + \overline{v}\frac{\partial \overline{v}}{\partial y} - \left(-\frac{1}{\rho}\right)\frac{\partial \overline{p}}{\partial y} - v\left(\frac{\partial^2 \overline{v}}{\partial x^2} + \frac{\partial^2 \overline{v}}{\partial y^2}\right) + \overline{v'\frac{\partial \overline{v}'}{\partial y}} + \overline{u'\frac{\partial \overline{v}'}{\partial x}} = 0 \tag{A24}$$

*Appendix A.3. Mixing Length Model*

The 2D stationary momentum equations read,

$$\overline{u}\frac{\partial \overline{u}}{\partial x} + \overline{v}\frac{\partial \overline{u}}{\partial y} - \left(-\frac{1}{\rho}\right)\frac{\partial \overline{p}}{\partial x} - v\left(\frac{\partial^2 \overline{u}}{\partial x^2} + \frac{\partial^2 \overline{u}}{\partial y^2}\right) + \overline{v'\frac{\partial \overline{u}'}{\partial y}} + \overline{u'\frac{\partial \overline{u}'}{\partial x}} = 0 \tag{A25}$$

$$\overline{u}\frac{\partial \overline{v}}{\partial x} + \overline{v}\frac{\partial \overline{v}}{\partial y} - \left(-\frac{1}{\rho}\right)\frac{\partial \overline{p}}{\partial y} - v\left(\frac{\partial^2 \overline{v}}{\partial x^2} + \frac{\partial^2 \overline{v}}{\partial y^2}\right) + \overline{v'\frac{\partial \overline{v}'}{\partial y}} + \overline{u'\frac{\partial \overline{v}'}{\partial x}} = 0$$

With the first-order stress terms given by,

$$\tau_{ii} = \overline{u'\frac{\partial u'}{\partial x}}, \tau_{ij} = \overline{u'\frac{\partial v'}{\partial x}}, tau_{ji} = \overline{v'\frac{\partial u'}{\partial x}}, \tau_{jj} = \overline{v'\frac{\partial v'}{\partial x}} \tag{A26}$$

According to the Boussinesq assumption, the Reynolds stresses can be given,

$$\tau_{ij} = \mu_t\left(2S_{ij} - \frac{2}{3}\frac{\partial u_k}{\partial x_k}\delta_{ij}\right) - \frac{2}{3}\rho k\delta_{ij} \tag{A27}$$

where, $\mu_t$ is a turbulent eddy viscosity and $S_{ij}$ is the stress tensor, given by,

$$S_{ij} = \frac{1}{2}\left(\frac{\partial u_i}{\partial x_j} + \frac{\partial u_j}{\partial x_i}\right) \tag{A28}$$

The time average of the stress tensor is given,

$$\overline{S_{ij}} = S_{ij} - \frac{1}{3}\frac{\partial u_k}{\partial x_k}\delta_{ij} \tag{A29}$$

Taking the time average of the Boussinesq assumption by applying the Reynolds decomposition,

$$u_k = \overline{u_k} + u'_k \tag{A30}$$

$$\overline{\tau_{ij}} = \mu_t\left(2\left(S_{ij} - \frac{1}{3}\frac{\partial(\overline{u_k} + u'_k)}{\partial x_k}\delta_{ij}\right) - \frac{2}{3}\frac{\partial(\overline{u_k} + u'_k)}{\partial x_k}\delta_{ij}\right) - \frac{2}{3}\rho k\delta_{ij} \tag{A31}$$

$$\overline{\tau_{ij}} = \mu_t\left(2(\overline{S_{ij}})\right) - \frac{2}{3}\rho k\delta_{ij} \tag{A32}$$

$$\overline{\tau_{ij}} = \mu_t \left( \left( \frac{\partial u_i}{\partial x_j} + \frac{\partial u_j}{\partial x_i} \right) \right) - \frac{2}{3} \rho k \delta_{ij} \tag{A33}$$

Since the time average of a time average is that same time average, and the equation must be written to deal with these time averages, this can be re-written,

$$\overline{\tau_{ij}} = \mu_t \left( \left( \frac{\partial \overline{u_i}}{\partial x_j} + \frac{\partial \overline{u_j}}{\partial x_i} \right) \right) - \frac{2}{3} \rho k \delta_{ij} \tag{A34}$$

Since the no turbulent kinetic energy, *k*, is calculated here, it is approximated,

$$k = \frac{1}{2} \overline{u_i' u_i'} = \frac{1}{2} (\overline{u' u'} + \overline{v' v'}) \tag{A35}$$

The first-order stress $\overline{\tau_{ii}}$ therefore reads,

$$\overline{\tau_{ii}} = \mu_t \left( \left( \frac{\partial \overline{u}}{\partial x} + \frac{\partial \overline{u}}{\partial x} \right) \right) - \frac{2}{3} \rho \left[ \frac{1}{2} (\overline{u' u'} + \overline{v' v'}) \right] \tag{A36}$$

where,

$$\overline{u' u'} = -\tau_{ii} \tag{A37}$$

$$\overline{v' v'} = -\tau_{jj} \tag{A38}$$

Therefore,

$$\overline{\tau_{ii}} = \mu_t \left( \frac{\partial \overline{u}}{\partial x} + \frac{\partial \overline{u}}{\partial x} \right) - \frac{2}{3} \rho \left[ \frac{1}{2} (\overline{-\tau_{ii}} + \overline{-\tau_{jj}}) \right] \tag{A39}$$

$$\overline{\tau_{ii}} = \mu_t \left( \frac{\partial \overline{u}}{\partial x} + \frac{\partial \overline{u}}{\partial x} \right) - \frac{1}{3} \rho \left[ (\overline{-\tau_{ii}} + \overline{-\tau_{jj}}) \right] \tag{A40}$$

The first-order stress $\tau_{jj}$ similarly reads,

$$\overline{\tau_{jj}} = \mu_t \left( \frac{\partial \overline{v}}{\partial y} + \frac{\partial \overline{v}}{\partial y} \right) - \frac{1}{3} \rho [(-\tau_{ii} + -\tau_{jj})] \tag{A41}$$

These equations can be reduced,

$$\overline{\tau_{ii}} \left( 1 - \frac{1}{3} \rho \right) = \overline{\tau_{jj}} \left( \frac{1}{3} \rho \right) + \mu_t \left( \frac{\partial \overline{u}}{\partial x} + \frac{\partial \overline{u}}{\partial x} \right) \tag{A42}$$

$$\overline{\tau_{jj}} \left( 1 - \frac{1}{3} \rho \right) = \overline{\tau_{ii}} \left( \frac{1}{3} \rho \right) + \mu_t \left( \frac{\partial \overline{v}}{\partial y} + \frac{\partial \overline{v}}{\partial y} \right) \tag{A43}$$

Therefore,

$$\overline{\tau_{ii}} = \frac{\overline{\tau_{jj}} \left( \frac{1}{3} \rho \right) + \mu_t \left( \frac{\partial \overline{u}}{\partial x} + \frac{\partial \overline{u}}{\partial x} \right)}{\left( 1 - \frac{1}{3} \rho \right)} \tag{A44}$$

$$\overline{\tau_{jj}} = \frac{\overline{\tau_{ii}} \left( \frac{1}{3} \rho \right) + \mu_t \left( \frac{\partial \overline{v}}{\partial y} + \frac{\partial \overline{v}}{\partial y} \right)}{\left( 1 - \frac{1}{3} \rho \right)} \tag{A45}$$

Substitution of $\overline{\tau_{jj}}$ into $\overline{\tau ii}$ gives,

$$\overline{\tau_{ii}} \left( 1 - \frac{1}{3} \rho \right) = - \frac{\overline{\tau_{ii}} \left( \frac{1}{3} \rho \right) + \mu_t \left( \frac{\partial \overline{v}}{\partial y} + \frac{\partial \overline{v}}{\partial y} \right)}{\left( 1 - \frac{1}{3} \rho \right)} \left( \frac{1}{3} \rho \right) + \mu_t \left( \frac{\partial \overline{u}}{\partial x} + \frac{\partial \overline{u}}{\partial x} \right) \tag{A46}$$

$$\overline{\tau_{ii}}\left(1 - \frac{1}{3}\rho\right)\left(1 - \frac{1}{3}\rho\right) = \overline{\tau_{ii}}\left(\frac{1}{3}\rho\right) + \mu_t\left(\frac{\partial\bar{v}}{\partial y} + \frac{\partial\bar{v}}{\partial y}\right)\left(\frac{1}{3}\rho\right) + \mu_t\left(\frac{\partial\bar{u}}{\partial x} + \frac{\partial\bar{u}}{\partial x}\right)\left(1 - \frac{1}{3}\rho\right) \tag{A47}$$

$$\overline{\tau_{ii}}\left(1 - \frac{1}{3}\rho\right)\left(1 - \frac{1}{3}\rho\right) - \overline{\tau_{ii}}\left(\frac{1}{3}\rho\right) = \mu_t\left(\frac{\partial\bar{v}}{\partial y} + \frac{\partial\bar{v}}{\partial y}\right)\left(\frac{1}{3}\rho\right) + \mu_t\left(\frac{\partial\bar{u}}{\partial x} + \frac{\partial\bar{u}}{\partial x}\right)\left(1 - \frac{1}{3}\rho\right) \tag{A48}$$

$$\overline{\tau_{ii}}\left(1 - \frac{1}{3}\rho\right)^2 - \overline{\tau_{ii}}\left(\frac{1}{3}\rho\right) = \mu_t\left(\frac{\partial\bar{v}}{\partial y} + \frac{\partial\bar{v}}{\partial y}\right)\left(\frac{1}{3}\rho\right) + \mu_t\left(\frac{\partial\bar{u}}{\partial x} + \frac{\partial\bar{u}}{\partial x}\right)\left(1 - \frac{1}{3}\rho\right) \tag{A49}$$

$$\overline{\tau_{ii}}\left(1 - \frac{2}{3}\rho - \frac{1}{9}\rho^2\right) - \overline{\tau_{ii}}\left(\frac{1}{3}\rho\right) = \mu_t\left(\frac{\partial\bar{v}}{\partial y} + \frac{\partial\bar{v}}{\partial y}\right)\left(\frac{1}{3}\rho\right) + \mu_t\left(\frac{\partial\bar{u}}{\partial x} + \frac{\partial\bar{u}}{\partial x}\right)\left(1 - \frac{1}{3}\rho\right) \tag{A50}$$

Since the density of water is around 1 g/m², 

$$\overline{\tau_{ii}}\left(1 - \frac{2}{3} - \frac{1}{9}\right) - \overline{\tau_{ii}}\left(\frac{1}{3}\right) = \mu_t\left(\frac{\partial\bar{v}}{\partial y} + \frac{\partial\bar{v}}{\partial y}\right)\left(\frac{1}{3}\right) + \mu_t\left(\frac{\partial\bar{u}}{\partial x} + \frac{\partial\bar{u}}{\partial x}\right)\left(1 - \frac{1}{3}\right) \tag{A51}$$

$$\frac{1}{3}\overline{\tau_{ii}} = \frac{1}{3}\mu_t\left(\frac{\partial\bar{v}}{\partial y} + \frac{\partial\bar{v}}{\partial y}\right) + \frac{2}{3}\mu_t\left(\frac{\partial\bar{u}}{\partial x} + \frac{\partial\bar{u}}{\partial x}\right) \tag{A52}$$

$$\overline{\tau_{ii}} = \mu_t\left(\frac{\partial\bar{v}}{\partial y} + \frac{\partial\bar{v}}{\partial y}\right) + 2\mu_t\left(\frac{\partial\bar{u}}{\partial x} + \frac{\partial\bar{u}}{\partial x}\right) \tag{A53}$$

$$\overline{\tau_{ii}} = 2\mu_t\left(\frac{\partial\bar{u}}{\partial x} + \frac{\partial\bar{u}}{\partial x} + \frac{\partial\bar{v}}{\partial y}\right) \tag{A54}$$

Similarly, the first-order stress for $\overline{\tau_{jj}}$ can be derived,

$$\overline{\tau_{jj}} = 2\mu_t\left(\frac{\partial\bar{u}}{\partial x} + \frac{\partial\bar{v}}{\partial y} + \frac{\partial\bar{v}}{\partial y}\right) \tag{A55}$$

The first-order stresses $\overline{\tau_{ii}}$ and $\overline{\tau_{jj}}$ do not require turbulent kinetic energy terms due to the Kronecker delta. As the fluid is incompressible,

$$\overline{\tau_{ij}} = \overline{\tau_{ji}} = \mu_t\left(\frac{\partial\bar{u}}{\partial y} + \frac{\partial\bar{v}}{\partial x}\right) \tag{A56}$$

The turbulent viscosity of the mixing length model is given based on the relation,

$$\mu_t = l_m^2\sqrt{G} \tag{A57}$$

where the mixing length, $l_m$, is given,

$$l_m = min(0.419d, 0.09d_{max}) \tag{A58}$$

where $d$ is the distance from the wall. The mean strain rate tensor, $G$, is given,

$$G = 2\left(\frac{\partial\bar{u}}{\partial x}\right)^2 + 2\left(\frac{\partial\bar{v}}{\partial y}\right)^2 + \left(\frac{\partial\bar{u}}{\partial y} + \frac{\partial\bar{v}}{\partial x}\right)^2 \tag{A59}$$

*Appendix A.4. K-Omega Model*

The general form of the turbulent kinetic energy, $k$, equation reads,

$$\frac{\partial(\rho k)}{\partial t} + \frac{\partial(\rho u_j k)}{\partial x_j} = P - \beta^*\rho\omega k + \frac{\partial}{\partial x_j}\left[\left(\mu + \sigma_k\frac{\rho k}{\omega}\right)\frac{\partial k}{\partial x_j}\right] \tag{A60}$$

where the production term is given,

$$P = \tau_{ij} \frac{\partial u_i}{\partial x_j} \tag{A61}$$

According to the Boussinesq assumption, the Reynolds stresses can be given,

$$\tau_{ij} = \mu_t \left( 2S_{ij} - \frac{2}{3} \frac{\partial u_k}{\partial x_k} \delta_{ij} \right) - \frac{2}{3} \rho k \delta_{ij} \tag{A62}$$

where $\mu_t$ is a turbulent eddy viscosity and $S_{ij}$ is the stress tensor, given by,

$$S_{ij} = \frac{1}{2} \left( \frac{\partial u_i}{\partial x_j} + \frac{\partial u_j}{\partial x_i} \right) \tag{A63}$$

The time average of the stress tensor,

$$\overline{S}_{ij} = S_{ij} - \frac{1}{3} \frac{\partial u_k}{\partial x_k} \delta_{ij} \tag{A64}$$

Taking the time average of the Boussinesq assumption by applying the Reynolds decomposition,

$$u_k = \overline{u_k} + u_k' \tag{A65}$$

$$\overline{\tau}_{ij} = \mu_t \left( 2 \left( S_{ij} - \frac{1}{3} \frac{\partial (\overline{u}_k + u_k')}{\partial x_k} \delta_{ij} \right) - \frac{2}{3} \frac{\partial (\overline{u}_k + u_k')}{\partial x_k} \delta_{ij} \right) - \frac{2}{3} \rho k \delta_{ij} \tag{A66}$$

$$\overline{\tau}_{ij} = \mu_t \left( 2 (\overline{S}_{ij}) \right) - \frac{2}{3} \rho k \delta_{ij} \tag{A67}$$

$$\overline{\tau}_{ij} = \mu_t \left( \left( \frac{\partial u_i}{\partial x_j} + \frac{\partial u_j}{\partial x_i} \right) \right) - \frac{2}{3} \rho k \delta_{ij} \tag{A68}$$

Since the time average of a time average is that same time average, and the equation must be written to deal with these time averages, this can be re-written,

$$\tau_{ij} = \mu_t \left( \frac{\partial \overline{u}_i}{\partial x_j} + \frac{\partial \overline{u}_j}{\partial x_i} \right) - \frac{2}{3} \rho k \delta_{ij} \tag{A69}$$

And the turbulent viscosity, $\mu_t$, is calculated,

$$\mu_t = \frac{\rho k}{\hat{\omega}} \tag{A70}$$

where,

$$\hat{\omega} = max \left( \omega, C_{lim} \sqrt{\frac{2 \overline{S_{ij} S_{ij}}}{\beta *}} \right) \; is \; simplified \; to \; \hat{\omega} = \omega \tag{A71}$$

This can be used to calculate the time average of the production term,

$$\overline{P} = \overline{\tau}_{ij} \frac{\partial \overline{u_i}}{\partial x_j} \tag{A72}$$

This can be used to calculate the time average of the k equation,

$$\frac{\partial (\rho \overline{k})}{\partial t} + \frac{\partial (\rho u_j \overline{k})}{\partial x_j} = \overline{P} - \beta^* \rho \omega \overline{k} + \frac{\partial}{\partial x_j} \left[ \left( \mu + \sigma_k \frac{\rho \overline{k}}{\omega} \right) \frac{\partial \overline{k}}{\partial x_j} \right] \tag{A73}$$

In this case, since $k$ and $\omega$ are unknowns, and their fluctuating components are not known or used, we can assume that the fluctuating components are zero, and hence leave them as they are in the equation. It should be noted that since $k$ is the turbulent kinetic energy of averaged quantities, therefore,

$$\frac{\partial \rho k}{\partial t} = 0 \tag{A74}$$

As $k$ does not change with time. This leaves,

$$\frac{\partial(\rho \bar{u}_j k)}{\partial x_j} = \bar{P} - \beta^* \rho \omega k + \frac{\partial}{\partial x_j}\left[\left(\mu + \sigma_k \frac{\rho k}{\omega}\right)\frac{\partial k}{\partial x_j}\right] \tag{A75}$$

Which can be expanded for the 2D stationary domain into,

$$\frac{\partial(\overline{\rho u k})}{\partial x} + \frac{\partial(\overline{\rho v k})}{\partial y} = \overline{\left(\tau_{xx}\frac{\partial \overline{u}}{\partial x} + \tau_{xy}\frac{\partial \overline{u}}{\partial y} + \tau_{yx}\frac{\partial \overline{v}}{\partial y} + \tau_{yy}\frac{\partial \overline{v}}{\partial y}\right)} - \beta^* \rho \omega k \tag{A76}$$

$$+ \frac{\partial}{\partial x}\left[\left(\mu + \sigma_k \frac{\rho k}{\omega}\right)\frac{\partial k}{\partial x}\right] + \frac{\partial}{\partial y}\left[\left(\mu + \sigma_k \frac{\rho k}{\omega}\right)\frac{\partial k}{\partial y}\right]$$

Using the product rule,

$$\frac{\partial(\overline{\rho u k})}{\partial x} + \frac{\partial(\overline{\rho v k})}{\partial y} = \rho\left(\left(\bar{u}\frac{\partial k}{\partial x} + k\frac{\partial \bar{u}}{\partial x}\right) + \left(\bar{v}\frac{\partial k}{\partial y} + k\frac{\partial \bar{v}}{\partial y}\right)\right) \tag{A77}$$

Giving,

$$\rho\left(\left(\bar{u}\frac{\partial k}{\partial x} + k\frac{\partial \bar{u}}{\partial x}\right) + \left(\bar{v}\frac{\partial k}{\partial y} + k\frac{\partial \bar{v}}{\partial y}\right)\right) = \left(\bar{\tau}_{xx}\frac{\partial \bar{u}}{\partial x} + \bar{\tau}_{xy}\frac{\partial \bar{u}}{\partial y} + \bar{\tau}_{yx}\frac{\partial \bar{v}}{\partial y} + \bar{\tau}_{yy}\frac{\partial \bar{v}}{\partial y}\right) - \beta^* \rho \omega k$$

$$+ \frac{\partial}{\partial x}\left[\left(\mu + \sigma_k \frac{\rho k}{\omega}\right)\frac{\partial k}{\partial x}\right] + \frac{\partial}{\partial y}\left[\left(\mu + \sigma_k \frac{\rho k}{\omega}\right)\frac{\partial k}{\partial y}\right] \tag{A78}$$

The general form of the turbulent kinetic energy dissipation, $\omega$, is given,

$$\frac{\partial(\rho \omega)}{\partial t} + \frac{\partial(\rho u_j \omega)}{\partial x_j} = \frac{\gamma \omega}{k}P - \beta \rho \omega^2 + \frac{\partial}{\partial x_j}\left[\left(\mu + \sigma_\omega \frac{\rho k}{\omega}\right)\frac{\partial \omega}{\partial x_j}\right] + \frac{\rho \sigma_d}{\omega}\frac{\partial k}{\partial x_j}\frac{\partial \omega}{\partial x_j} \tag{A79}$$

Which can be similarly rewritten,

$$\rho\left(\left(\bar{u}\frac{\partial \omega}{\partial x} + \omega\frac{\partial \bar{u}}{\partial x}\right) + \left(\bar{v}\frac{\partial \omega}{\partial y} + \omega\frac{\partial \bar{v}}{\partial y}\right)\right) = \frac{\gamma \omega}{k}\left(\bar{\tau}_{xx}\frac{\partial \bar{u}}{\partial x} + \bar{\tau}_{xy}\frac{\partial \bar{u}}{\partial y} + \bar{\tau}_{yx}\frac{\partial \bar{v}}{\partial y} + \bar{\tau}_{yy}\frac{\partial \bar{v}}{\partial y}\right) - \beta \rho \omega^2$$

$$+ \frac{\partial}{\partial x}\left[\left(\mu + \sigma_\omega \frac{\rho k}{\omega}\right)\frac{\partial \omega}{\partial x}\right] + \frac{\rho \sigma_d}{\omega}\frac{\partial k}{\partial x}\frac{\partial \omega}{\partial x} + \frac{\partial}{\partial y}\left[\left(\mu + \sigma_\omega \frac{\rho k}{\omega}\right)\frac{\partial \omega}{\partial y}\right] + \frac{\rho \sigma_d}{\omega}\frac{\partial k}{\partial y}\frac{\partial \omega}{\partial y} \tag{A80}$$

**Appendix B**

Equations for the geometry, as provided in [19]

$\hat{y} = min(1 : 1 + 2.42 \times 10^{-4}\hat{x}^2 - 7.588 \times 10^5\hat{x}^3), \hat{x} \in [0, 0.3214]$

$\hat{y} = 0.8955 + 3.4844\hat{x} - 3.629 \times 10^{-3}\hat{x}^2 + 6.749 \times 10^{-5}\hat{x}^3, \hat{x} \in [0.3214, 0.5]$

$\hat{y} = 0.9213 + 2.931 \times 10^{-2}\hat{x} + 3.243 \times 10^{-3}\hat{x}^2 + 5.809 \times 10^{-5}\hat{x}^3, \hat{x} \in [0.5, 0.7143]$

$\hat{y} = 1.445 - 4.927 \times 10^{-2}\hat{x} + 6.95 \times 10^{-4}\hat{x}^2 - 7.394 \times 10^{-6}\hat{x}^3, \hat{x} \in [0.7143, 1.071]$

$\hat{y} = 0.6401 + 3.1123 \times 10^{-2}\hat{x} + 1.988 \times 10^{-3}\hat{x}^2 + 2.242 \times 10^{-6}\hat{x}^3, \hat{x} \in [1.071, 1.429]$

$\hat{y} = max(0 : 2.0139 - 7.18 \times 10^{-4}\hat{x} + 5.875 \times 10^{-4}\hat{x}^2 + 9.553 \times 10^{-7}\hat{x}^3,) \hat{x} \in [1.1429, 1.929]$

where $\hat{x} = \frac{x}{H}$ and $\hat{y} = \frac{y}{H}$ are normalised horizontal and vertical coordinates, respectively.

## Appendix C

Neural networks are used to model relationships between independent variables such as geometry and dependant variables such as flow fields and stress terms in the case of turbulence modelling. When referring to a neural network that makes an exact prediction rather than classification, this is known as regression.

The simplest form of neural networks used for regression are dense multilayer perceptron (MLP) or feed-forward neural networks. A dense MLP network is characterised by its unidirectional information flow from input nodes to a set of interconnected nodes that are weighted to transform data as they are passed from one node to another.

The neural network can be conceptually simplified into a simple polynomial approximation, where the hidden nodes represent the equation relating the independent variables to the dependent variables, in the same way a simple transformer might behave. The training of the neural network is simply the method of determining the correct nodal weights to correctly characterise the transformer, to ensure that it is representative of the correct nonlinear dynamics between the independent and dependent variables. This architecture is represented in a general form in Figure A1. The method of training is via calculation of some loss, as discussed in the Methods section. Backpropagation is used for optimising this loss function by adjusting weights according to gradients calculated via chain rule-based computation on losses relative to node weights and biases resulting from the backpropagation method. These gradients are then employed within an optimiser such as Adam, which uses them iteratively to recalibrate node weights, ensuring convergence towards accurate solutions.

Once the neural network has been trained, it can predict the value of independent variables from new dependant variables, and, provided that the weights have adjusted sufficiently to capture the nonlinear dynamics of the data, the predictions should be accurate and representative of the test data.

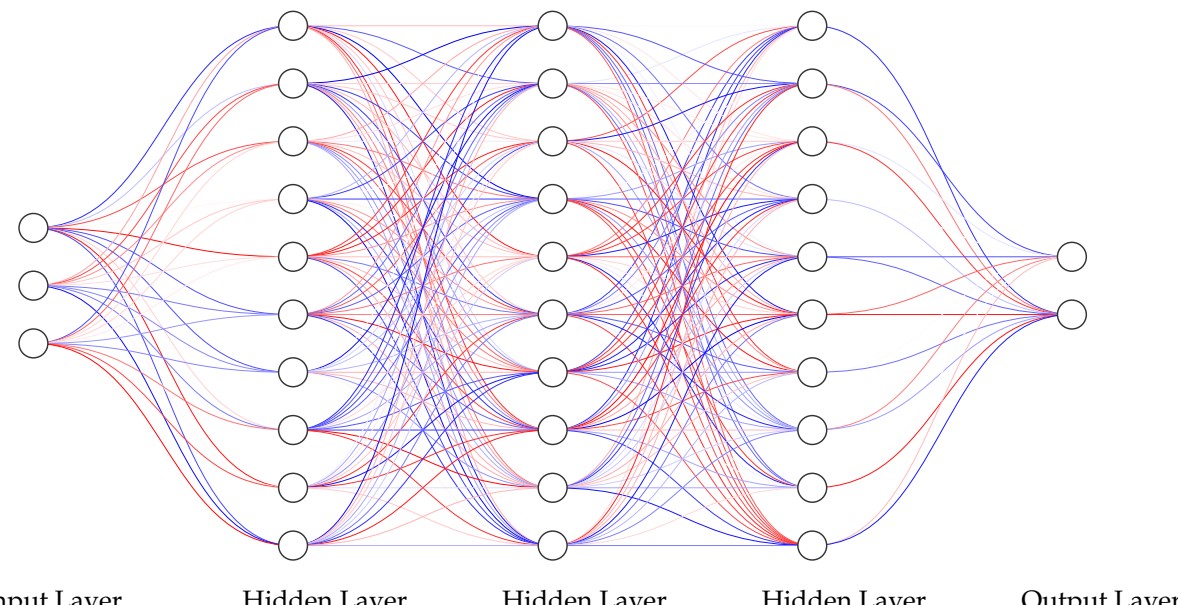

| Input Layer | Hidden Layer | Hidden Layer | Hidden Layer | Output Layer |

**Figure A1.** General dense MLP neural network architecture.

## Appendix D

*Appendix D.1. Methods*

The k omega model is a two-equation turbulence model that defines mean flow in terms of turbulence kinetic energy, *k*, which is closed using the kinetic energy dissipation rate, $\omega$. The 2D stationary conservation equations are written,

$$\frac{\partial(\rho k)}{\partial t} + \frac{\partial(\rho u_j k)}{\partial x_j} = P - \beta^* \rho \omega k + \frac{\partial}{\partial x_j}\left[\left(\mu + \sigma_k \frac{\rho k}{\omega}\right)\frac{\partial k}{\partial x_j}\right] \tag{A81}$$

$$\frac{\partial(\rho \omega)}{\partial t} + \frac{\partial(\rho u_j \omega)}{\partial x_j} = \frac{\gamma \omega}{k}P - \beta \rho \omega^2 + \frac{\partial}{\partial x_j}\left[\left(\mu + \sigma_\omega \frac{\rho k}{\omega}\right)\frac{\partial \omega}{\partial x_j}\right] + \frac{\rho \sigma_d}{\omega}\frac{\partial k}{\partial x_j}\frac{\partial \omega}{\partial x_j} \tag{A82}$$

These can be simplified and rewritten in a singular partial differential form for mean flow,

$$\rho\left(\left(\bar{u}\frac{\partial k}{\partial x} + k\frac{\partial \bar{u}}{\partial x}\right) + \left(\bar{v}\frac{\partial k}{\partial y} + k\frac{\partial \bar{v}}{\partial y}\right)\right) = \left(\bar{\tau}_{xx}\frac{\partial \bar{u}}{\partial x} + \bar{\tau}_{xy}\frac{\partial \bar{u}}{\partial y} + \bar{\tau}_{yx}\frac{\partial \bar{v}}{\partial y} + \bar{\tau}_{yy}\frac{\partial \bar{v}}{\partial y}\right) - \beta^* \rho \omega k$$
$$+ \frac{\partial}{\partial x}\left[\left(\mu + \sigma_k \frac{\rho k}{\omega}\right)\frac{\partial k}{\partial x}\right] + \frac{\partial}{\partial y}\left[\left(\mu + \sigma_k \frac{\rho k}{\omega}\right)\frac{\partial k}{\partial y}\right] \tag{A83}$$

$$\rho\left(\left(\bar{u}\frac{\partial \omega}{\partial x} + \omega\frac{\partial \bar{u}}{\partial x}\right) + \left(\bar{v}\frac{\partial \omega}{\partial y} + \omega\frac{\partial \bar{v}}{\partial y}\right)\right) = \frac{\gamma \omega}{k}\left(\bar{\tau}_{xx}\frac{\partial \bar{u}}{\partial x} + \bar{\tau}_{xy}\frac{\partial \bar{u}}{\partial y} + \bar{\tau}_{yx}\frac{\partial \bar{v}}{\partial y} + \bar{\tau}_{yy}\frac{\partial \bar{v}}{\partial y}\right) - \beta \rho \omega^2$$
$$+ \frac{\partial}{\partial x}\left[\left(\mu + \sigma_\omega \frac{\rho k}{\omega}\right)\frac{\partial \omega}{\partial x}\right] + \frac{\rho \sigma_d}{\omega}\frac{\partial k}{\partial x}\frac{\partial \omega}{\partial x} + \frac{\partial}{\partial y}\left[\left(\mu + \sigma_\omega \frac{\rho k}{\omega}\right)\frac{\partial \omega}{\partial y}\right] + \frac{\rho \sigma_d}{\omega}\frac{\partial k}{\partial y}\frac{\partial \omega}{\partial y} \tag{A84}$$

Turbulent viscosity for this model is calculated,

$$\mu_t = \frac{\rho k}{\hat{\omega}} \tag{A85}$$

where,

$$\hat{\omega} = max\left(\omega, C_{lim}\sqrt{\frac{2\overline{S_{ij}S_{ij}}}{\beta *}}\right) \; is \; simplified \; to \; \hat{\omega} = \omega \tag{A86}$$

A full derivation can be found in Appendix A. See Table A1 for full constraint list.

*Appendix D.2. KOM Model*

**Table A1.** KOM model summary.

| Method Name | Neural Network Inputs | Neural Network Outputs | Governing Flow Equations | Enforced Boundaries |
|---|---|---|---|---|
| K-Omega | $\begin{pmatrix} x \\ y \end{pmatrix}$ | $\begin{pmatrix} \bar{u} \\ \bar{v} \\ k \\ \omega \end{pmatrix}$ | 2D turbulent kinetic energy stationary: $\rho\left(\left(\bar{u}\frac{\partial k}{\partial x} + k\frac{\partial \bar{u}}{\partial x}\right) + \left(\bar{v}\frac{\partial k}{\partial y} + k\frac{\partial \bar{v}}{\partial y}\right)\right) = \left(\bar{\tau}_{xx}\frac{\partial \bar{u}}{\partial x} + \bar{\tau}_{xy}\frac{\partial \bar{u}}{\partial y} + \bar{\tau}_{yx}\frac{\partial \bar{v}}{\partial y} + \bar{\tau}_{yy}\frac{\partial \bar{v}}{\partial y}\right) - \beta^* \rho \omega k + \frac{\partial}{\partial x}\left[\left(\mu + \sigma_k \frac{\rho k}{\omega}\right)\frac{\partial k}{\partial x}\right] + \frac{\partial}{\partial y}\left[\left(\mu + \sigma_k \frac{\rho k}{\omega}\right)\frac{\partial k}{\partial y}\right]$ 2D turbulent kinetic energy dissipation stationary: $\rho\left(\left(\bar{u}\frac{\partial \omega}{\partial x} + \omega\frac{\partial \bar{u}}{\partial x}\right) + \left(\bar{v}\frac{\partial \omega}{\partial y} + \omega\frac{\partial \bar{v}}{\partial y}\right)\right) = \frac{\gamma \omega}{k}\left(\bar{\tau}_{xx}\frac{\partial \bar{u}}{\partial x} + \bar{\tau}_{xy}\frac{\partial \bar{u}}{\partial y} + \bar{\tau}_{yx}\frac{\partial \bar{v}}{\partial y} + \bar{\tau}_{yy}\frac{\partial \bar{v}}{\partial y}\right) - \beta \rho \omega^2 + \frac{\partial}{\partial x}\left[\left(\mu + \sigma_\omega \frac{\rho k}{\omega}\right)\frac{\partial \omega}{\partial x}\right] + \frac{\rho \sigma_d}{\omega}\frac{\partial k}{\partial x}\frac{\partial \omega}{\partial x} + \frac{\partial}{\partial y}\left[\left(\mu + \sigma_\omega \frac{\rho k}{\omega}\right)\frac{\partial \omega}{\partial y}\right] + \frac{\rho \sigma_d}{\omega}\frac{\partial k}{\partial y}\frac{\partial \omega}{\partial y}$ Turbulent viscosity: $\mu_t = \frac{\rho k}{\hat{\omega}}$ Turbulent kinetic energy dissipation rate limit: $\omega = \hat{\omega}$ First-order stresses: $\overline{\tau_{xx}} = \mu_t\left(\frac{\partial \bar{u}}{\partial x} + \frac{\partial \bar{u}}{\partial x}\right) - \frac{2}{3}\rho k$ $\overline{\tau_{yy}} = \mu_t\left(\frac{\partial \bar{u}}{\partial y} + \frac{\partial \bar{v}}{\partial x}\right)$ $\overline{\tau_{xx}} = \mu_t\left(\frac{\partial \bar{v}}{\partial y} + \frac{\partial \bar{v}}{\partial y}\right) - \frac{2}{3}\rho k$ | $\begin{pmatrix} \bar{u} \\ \bar{v} \end{pmatrix}$ |

**Table A2.** Solver metrics and overall accuracy.

| Neural Network Name | Neural Network Abbreviation | Training Time (s) | Prediction Time | $\overline{u}$ Error (%) * | $\overline{v}$ Error (%) * | $\overline{p}$ Error (%) * | Description |
|---|---|---|---|---|---|---|---|
| K-Omega Model | KOM | 1.83 | - | 0.161 | 2.28 | - | Learning Rate of $1 \times 10^{-2}$ required for convergence, compared with $1 \times 10^{-3}$ for rest. Compared with others, time should be slower than shown. |

* See Appendix E for calculation method.

*Appendix D.3. KOM Results*

The KOM is not entirely comparable with the other results, as the KOM model failed to converge with the same solver parameters. To allow the model to solve, the learning rate was increased to $1 \times 10^{-2}$ from $1 \times 10^{-3}$ to provide faster initial convergence. The model has low error, close to the accuracy of the COM model, with a streamwise error of 0.161% and transverse velocity error of 2.28%. There are no pressure predictions for the KOM model, as pressure is not a variable present in the governing equations. The solve time is still much higher than the COM at 1.83 h, despite the learning rate decreasing the solve time.

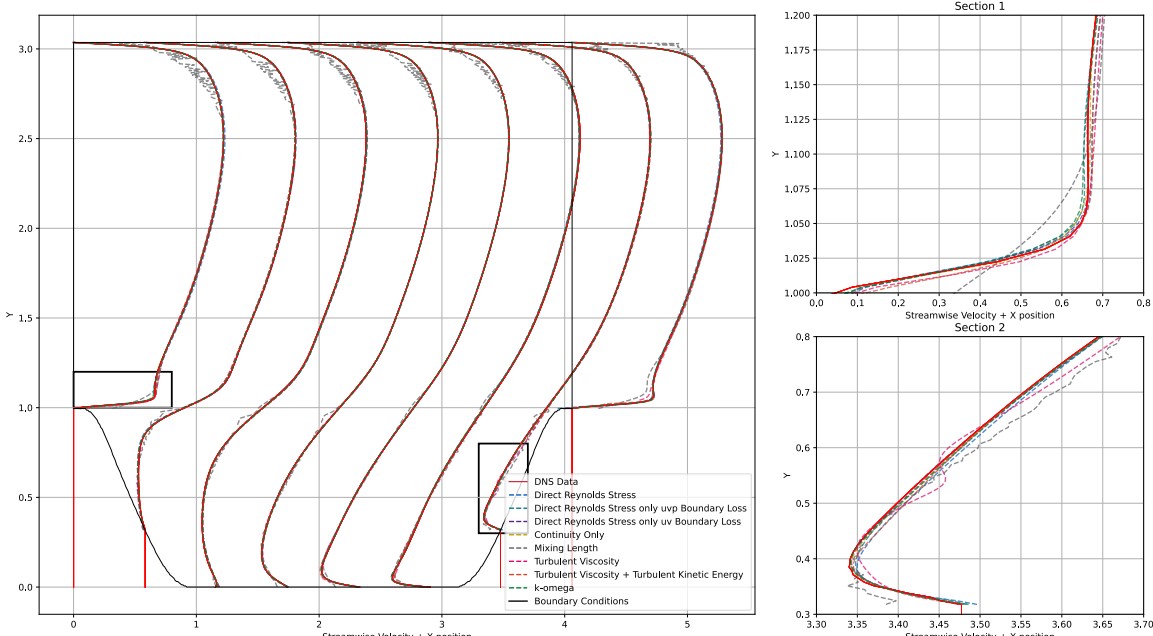

**Figure A2.** Time-averaged streamwise velocity, $\overline{u}$, PINNs compared with KOM.

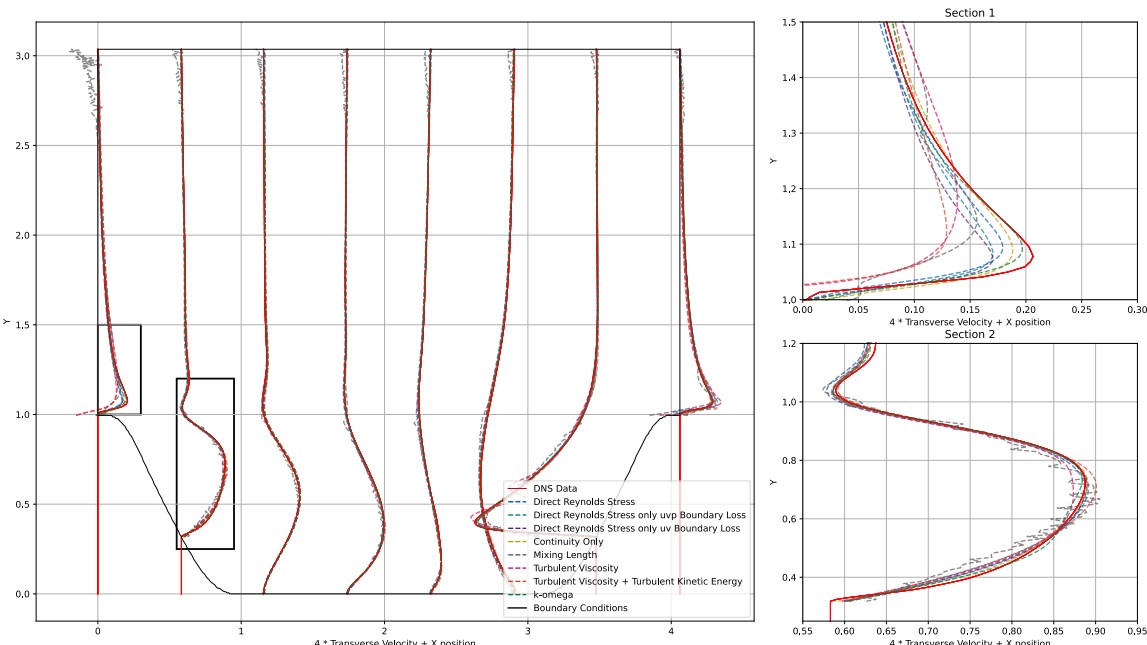

**Figure A3.** Time-averaged transverse velocity, $\overline{v}$, PINNs compared with KOM.

*Appendix D.4. KOM Discussion*

The KOM results show that the continuity form of the two equation models is sufficient to govern the model, and the results shown are fairly accurate. However, it is noted that the number of unknowns in each equation is higher than any other model, and governing this many unknowns using only two equations is likely the cause of the inability to converge within the same threshold using the same small learning rate as the other models. While this means the model is not directly comparable, this model is promising, avoiding the poor assumptions as in the MLM while directly predicting k and omega. In this manner, the number of equations being solved can be reduced from the TVKEM model, but the convergence threshold or learning rate will need to be adjusted to produce results. This shows that this form is accurate, but highlights the limitations of under-constrained models with a low number of governing equations and a high number of variables across them, with convergence issues being the most important.

**Appendix E**

The formula used to calculate the overall error per variable in the fluid domain is given,

$$\text{Total Domain Error (\%)} = \frac{\|\text{ref} - \text{pred}\|_{2,(1,2)}}{\|\text{ref}\|_{2,(1,2)}} \times 100 \tag{A87}$$

where ref and pred are the lists of reference and predicted fluid domain values, and $\|a\|_{2,(1,2)}$ denotes the values on the specified axis, which represent the points and variables.

It should be noted that there are several limitations with this method of calculating average domain error. This function is inherently sensitive to outliers. If there are extreme differences in the percentage error between the reference and predicted points in the domain, this can disproportionately affect the error calculation. This can lead to large errors being cancelled, skewed error, and other effects leading to a lower or higher overall percentage error than might accurately represent the flow.

## Appendix F

**Table A3.** Data-driven metrics for all geometries. Runs 1, 2, and 3 solve time and epochs.

| Testing Geometry (Filename) | Run 1 Time (hrs) | Run 2 Time (hrs) | Run 3 Time (hrs) | Run 1 Epochs | Run 2 Epochs | Run 3 Epochs |
|---|---|---|---|---|---|---|
| 05-10071-2024 | 4.1465 | 4.8044 | 2.3025 | 64 | 75 | 36 |
| 05-10071-3036 | 4.2917 | 6.0054 | 3.6440 | 71 | 100 | 60 |
| 05-10071-4048 | 6.4153 | 4.5389 | 3.6440 | 100 | 72 | 58 |
| 05-4071-2024 | 6.5736 | 6.5935 | 6.5516 | 100 | 100 | 100 |
| 05-4071-3036 | 3.1383 | 3.3755 | 2.1277 | 51 | 54 | 34 |
| 05-4071-4048 | 6.4042 | 3.8653 | 4.8055 | 98 | 58 | 75 |
| 05-7071-2024 | 6.1510 | 6.2457 | 6.1696 | 100 | 100 | 100 |
| 05-7071-3036 | 6.0619 | 2.7623 | 5.9172 | 100 | 44 | 95 |
| 05-7071-4048 | 6.0049 | 6.1623 | 4.4935 | 98 | 100 | 73 |
| 075-80355-3036 | 2.1614 | 2.0215 | 3.5191 | 35 | 32 | 57 |
| 10-12-2024 | 4.0714 | 2.6690 | 6.3826 | 62 | 41 | 100 |
| 10-12-3036 | 3.3554 | 6.1844 | 3.9754 | 55 | 100 | 66 |
| 10-12-4048 | 3.9153 | 3.6145 | 2.6540 | 62 | 57 | 43 |
| 10-6-2024 | 1.9820 | 6.6085 | 6.2423 | 31 | 100 | 94 |
| 10-6-3036 | 4.1323 | 4.1945 | 6.1153 | 67 | 68 | 100 |
| 10-6-4048 | 2.5583 | 4.5273 | 4.4232 | 40 | 70 | 69 |
| 10-9-2024 | 2.9653 | 6.6347 | 2.7177 | 46 | 100 | 41 |
| 10-9-3036 | 2.1101 | 2.4105 | 2.9650 | 35 | 39 | 48 |
| 10-9-4048 | 3.6485 | 2.6676 | 6.5280 | 59 | 42 | 98 |
| 125-99645-3036 | 6.1283 | 6.2453 | 6.0690 | 100 | 100 | 100 |
| 15-10929-2024 | 4.5933 | 4.5386 | 4.1385 | 74 | 72 | 67 |
| 15-10929-3036 | 1.8833 | 6.0567 | 1.9385 | 31 | 100 | 32 |
| 15-10929-4048 | 2.3954 | 2.1975 | 2.0831 | 40 | 36 | 35 |
| 15-13929-2024 | 3.7073 | 2.3868 | 2.1093 | 59 | 37 | 33 |
| 15-13929-3036 | 2.9063 | 2.2772 | 2.6044 | 48 | 38 | 44 |
| 15-13929-4048 | 1.9373 | 2.0007 | 1.9445 | 31 | 32 | 31 |
| 15-7929-2024 | 2.2224 | 3.0369 | 3.5955 | 34 | 46 | 55 |
| 15-7929-3036 | 2.3923 | 1.9600 | 3.7755 | 39 | 32 | 61 |
| 15-7929-4048 | 3.9381 | 6.1410 | 3.5175 | 61 | 96 | 55 |
| Median Average | 3.6485 | 2.2772 | 3.7755 | 59 | 69 | 61 |

**Table A4.** Data-driven metrics for all geometries. Green is the geometry used for PINNs initial testing. Red show highest error (anomalous). Runs 1, 2, and 3 errors.

| Testing Geometry (Filename) | $\bar{u}$ Error (%) | | | $\bar{v}$ Error (%) | | | $\bar{p}$ Error (%) | | |
|---|---|---|---|---|---|---|---|---|---|
| | Run 1 | Run 2 | Run 3 | Run 1 | Run 2 | Run 3 | Run 1 | Run 2 | Run 3 |
| 05-10071-2024 | 10.7 | 8.1 | 7.4 | 53.6 | 39.9 | 36.0 | 107.4 | 108.3 | 107.2 |
| 05-10071-3036 | 3.8 | 2.4 | 3.6 | 25.3 | 16.9 | 19.6 | 109.1 | 111.3 | 110.9 |
| 05-10071-4048 | 2.4 | 2.4 | 3.9 | 23.7 | 15.8 | 30.9 | 117.2 | 115.9 | 114.7 |
| 05-4071-2024 | 9.1 | 10.0 | 10.1 | 69.3 | 61.1 | 68.4 | 157.6 | 166.3 | 146.5 |
| 05-4071-3036 | 2.8 | 4.1 | 3.5 | 67.5 | 66.6 | 64.0 | 213.9 | 222.6 | 215.8 |
| 05-4071-4048 | 2.9 | 3.5 | 6.1 | 60.7 | 64.7 | 84.8 | 253.9 | 241.9 | 251.6 |
| 05-7071-2024 | 5.3 | 5.9 | 6.0 | 39.6 | 36.7 | 44.6 | 110.4 | 110.1 | 111.9 |
| 05-7071-3036 | 2.9 | 3.4 | 2.5 | 30.0 | 33.3 | 23.6 | 123.3 | 125.1 | 122.3 |
| 05-7071-4048 | 2.2 | 1.8 | 1.9 | 23.8 | 24.9 | 23.8 | 135.1 | 133.4 | 130.5 |
| 075-80355-3036 | 19.2 | 15.1 | 13.1 | 127.6 | 109.5 | 94.0 | 130.5 | 127.6 | 112.5 |
| 10-12-2024 | 8226.0 | 33.7 | 34.2 | 5213.0 | 113.8 | 116.5 | 7328.0 | 111.4 | 111.8 |
| 10-12-3036 | 19.3 | 15.6 | 14.9 | 77.7 | 86.9 | 61.3 | 113.0 | 110.9 | 111.8 |
| 10-12-4048 | 18.4 | 15.0 | 18.1 | 103.6 | 84.4 | 92.2 | 114.7 | 113.2 | 113.7 |
| 10-6-2024 | 17.4 | 16.9 | 15.6 | 117.0 | 114.0 | 112.5 | 118.2 | 117.7 | 118.0 |
| 10-6-3036 | 9.8 | 6.2 | 9.7 | 95.4 | 60.9 | 101.4 | 147.0 | 146.9 | 147.4 |
| 10-6-4048 | 10.0 | 9.4 | 9.3 | 146.4 | 133.7 | 135.0 | 174.7 | 175.0 | 178.9 |
| 10-9-2024 | 21.0 | 21.4 | 21.0 | 81.3 | 79.8 | 94.3 | 106.6 | 108.5 | 108.6 |
| 10-9-3036 | 14.8 | 14.0 | 10.0 | 90.0 | 88.4 | 49.9 | 114.9 | 114.0 | 114.6 |
| 10-9-4048 | 10.6 | 9.9 | 6.3 | 83.4 | 81.3 | 56.7 | 119.9 | 120.9 | 119.1 |
| 125-99645-3036 | 1169.5 | 34.7 | 28.6 | 1431.7 | 110.7 | 126.6 | 1372.3 | 96.5 | 102.4 |
| 15-10929-2024 | 12.0 | 10.1 | 10.4 | 39.2 | 33.9 | 39.0 | 103.6 | 103.6 | 103.3 |
| 15-10929-3036 | 18.4 | 5.0 | 4.6 | 87.3 | 29.1 | 26.0 | 106.3 | 106.9 | 106.3 |
| 15-10929-4048 | 5.5 | 5.4 | 5.3 | 30.2 | 26.4 | 31.6 | 106.6 | 107.5 | 108.5 |
| 15-13929-2024 | 31.4 | 7519.2 | 146,994.1 | 109.9 | 86,986.4 | 135,865.8 | 108.9 | 28,905.0 | 417,257.2 |
| 15-13929-3036 | 25.2 | 27.9 | 20.0 | 95.5 | 105.6 | 110.7 | 108.4 | 109.9 | 106.4 |
| 15-13929-4048 | 30.7 | 20.8 | 120.3 | 103.2 | 105.5 | 6150.2 | 110.5 | 110.7 | 184.9 |
| 15-7929-2024 | 25.4 | 25.5 | 23.6 | 98.6 | 98.6 | 91.0 | 104.4 | 104.7 | 105.4 |
| 15-7929-3036 | 17.7 | 18.9 | 18.0 | 97.9 | 97.8 | 99.0 | 111.8 | 110.1 | 111.6 |
| 15-7929-4048 | 14.0 | 14.3 | 14.9 | 110.3 | 94.2 | 98.7 | 119.5 | 118.3 | 116.7 |
| Median Average | 12.0 | 10.0 | 10.0 | 83.4 | 80.5 | 68.4 | 114.7 | 112.3 | 112.5 |

## Appendix G

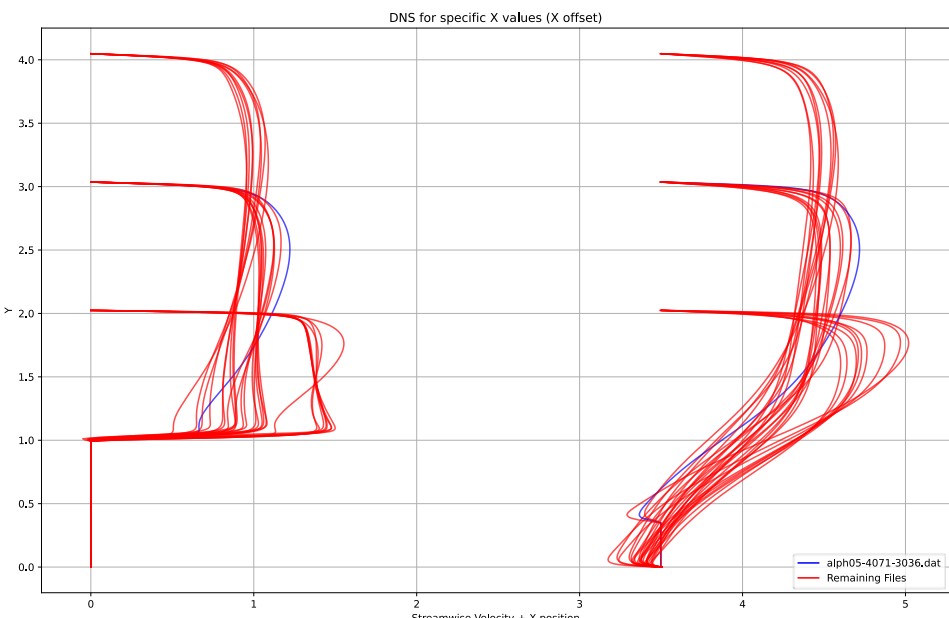

**Figure A4.** DNS compared for steamwise velocity at x = 0, x = 3.

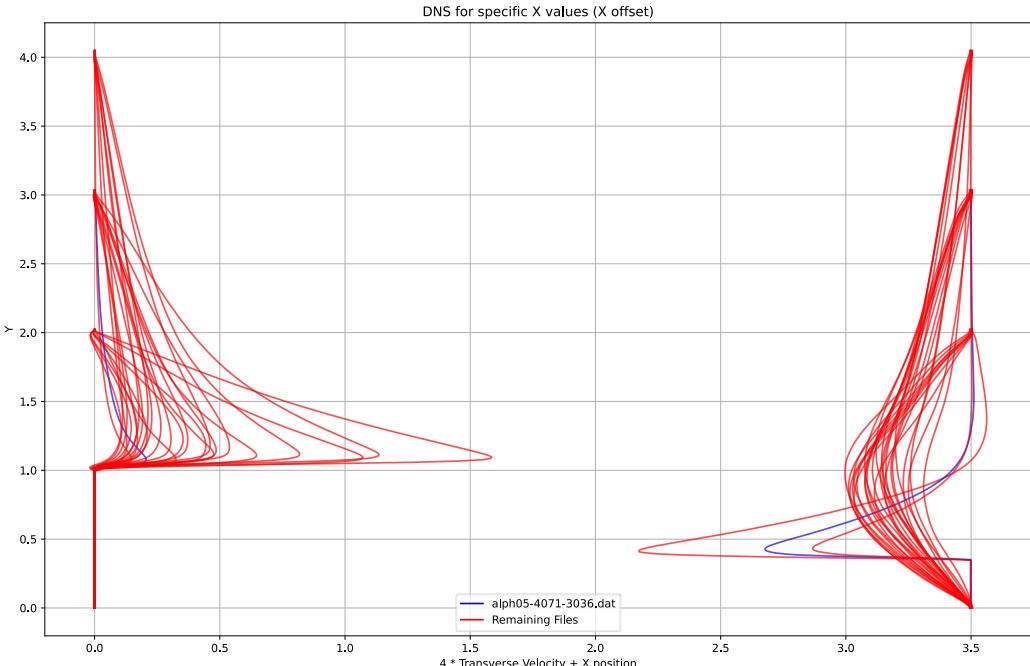

**Figure A5.** DNS compared for transverse velocity at x = 0, x = 3.

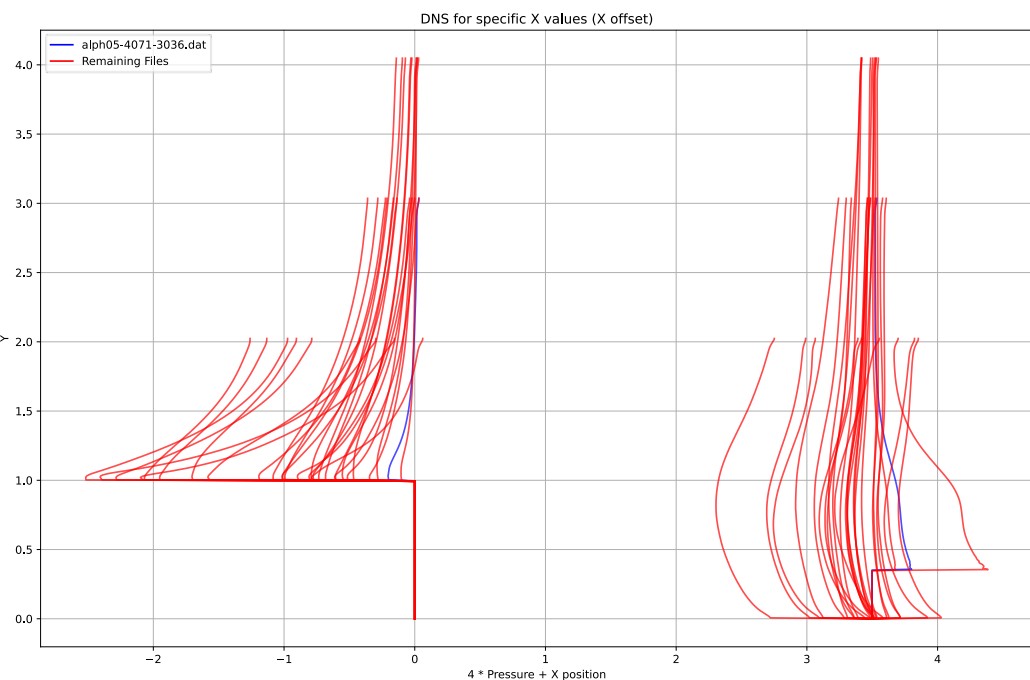

**Figure A6.** DNS compared for pressure at x = 0, x = 3.

**Appendix H**

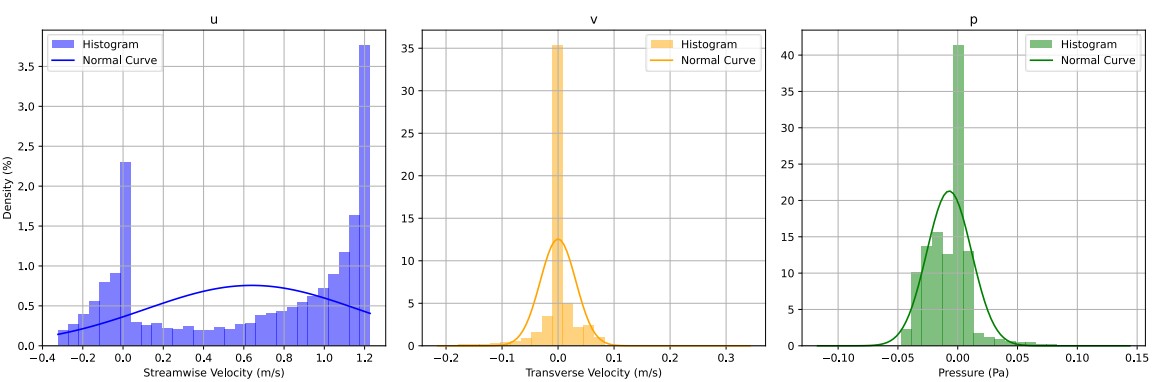

**Figure A7.** Data distribution for alph05-4071-3036.dat.

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
