# Peer review of "Optimising Physics-Informed Neural Network Solvers for Turbulence Modelling: A Study on Solver Constraints Against a Data-Driven Approach"

_fluids, doi:10.3390/fluids9120279_

Round 1
Reviewer 1 Report
Comments and Suggestions for Authors
My opinion is that the author did not read any book on turbulence. This is clear from the appendix. The paper therefore can not be published.
Comments on the Quality of English LanguageSome of the sentences do not have any meaning.
Reviewer 2 Report
Comments and Suggestions for Authors
This study examines solver constraints in a PINNs solver, aiming to generate an understanding of an optimal PINNs solver with reduced constraints compared to the numerically closed models used in traditional computational fluid dynamics (CFD).
The paper is written well and it looks interesting. The introduction needs to be improved a lot. You mentioned that PINNs has been used widely. You mentioned the disadvantages. However, you have not mentioned the improvement methods to PINNs such as Physics-data combined machine learning method, which has been used for parametric reduced-order modelling of nonlinear dynamical systems in small-data regimes. It is presented for improving extrapolation ability even using small training data.
Also, you mentioned reduced order models in the introduction, however, you do not mention any reduced order model(ROM) papers, in particular data driven reduced order model such as non-intrusive reduced order model constructed via RBF interpolation and self-attention.
Comments on the Quality of English LanguageN/A
Reviewer 3 Report
Comments and Suggestions for Authors
The current version of the article is very well presented. Indeed, the formulation of the problem and the methodology for its resolution are very clear. Furthermore, the quality of the figures and tables is very good, and the presentation of the equations as an appendix is judicious. Therefore, I recommend the publication of this article referenced as "fluids-3231533" after considering the following minor modifications:
1. Please integrate the numbering of all your equations within the main text.
2. Kindly mention the two parameters “\(\alpha\)” and “l” in Figure 1.
3. Please correct the typographical error (line 278).
4. Please specify the "solve time" in Table 5 in hours, as it is more quantifiable for this type of investigation. The same remark applies to Tables 6 and A3 in Appendix F.
5. Kindly mention the publication date of reference 2 in the references section.
6. Please enrich and update your state of the art in the "discussion" section by integrating more references (published between 2020-2024) relevant to the objective of your modeling study, given that the references section does not present enough references published between 2020-2024.
Reviewer 4 Report
Comments and Suggestions for Authors
Dear Editor
Concerns MS entitled: Optimising Physics Informed Neural Network Solvers for Turbulence Modelling: A Study on Solver Constraints against a Data Driven Approach
Generally, MS is interested and good written, also, results is good and suitable for the phenomenon studied, but needs some modifications before considering for publication in Fluid as follows:
- Abstract must summarized and comparison between the current results and previous obtained by others.
-In Keywords "Fluid Dynamics;" must be deleted.
-Introduction section is too poor and must be updated with adding more contributions.
-Equations 1-6 "A difficulty in turbulence modelling" must but in a new section "Formulation of the problem" not in introduction.
-Equations 7 and 8 must be proof or cited.
-Line 229 , "Equation 1, Equation 2, and Equation 3" modify to "Equations 1-3".
-It is worth to separate all symbols to "Nomenclature".
-Eqs. 20 and 21 do not correct written, must be revised, How obtained from Eqs. 15 and 19? authors must proof that.
-Figure 2 needs more details.
-All figures need to put in more resolution form.
-Conclusion doesn't enough, more details with physical meaning and the new results obtained needed to show the applications of Neural Network Solvers for Turbulence Modelling on the phenomenon.
-In Appendix A, there are some equations repeat in text and Appendices, no need to repeat.
Comments on the Quality of English LanguageEnglish language must be revised.
Round 2
Reviewer 4 Report
Comments and Suggestions for Authors
The authors made the requested queries, so, I recommend it for publication in its current form.
Sincerely Yours